# Learning with Calibration: Exploring Test-Time Computing of Spatio-Temporal Forecasting

**Wei Chen,  Yuxuan Liang**[*]
INTR & DSA Thrust, The Hong Kong University of Science and Technology (Guangzhou)
`onedeanxxx@gmail.com, yuxliang@outlook.com`

## Abstract

Spatio-temporal forecasting is crucial in many domains, such as transportation, meteorology, and energy. However, real-world scenarios frequently present challenges such as signal anomalies, noise, and distributional shifts. Existing solutions primarily enhance robustness by modifying network architectures or training procedures. Nevertheless, these approaches are computationally intensive and resource-demanding, especially for large-scale applications. In this paper, we explore a *novel test-time computing paradigm, namely learning with calibration, ST-TTC, for spatio-temporal forecasting*. Through learning with calibration, we aim to capture periodic structural biases arising from non-stationarity during the testing phase and perform real-time bias correction on predictions to improve accuracy. Specifically, we first introduce a spectral-domain calibrator with phase-amplitude modulation to mitigate periodic shift and then propose a flash updating mechanism with a streaming memory queue for efficient test-time computation. ST-TTC effectively bypasses complex training-stage techniques, offering an efficient and generalizable paradigm. Extensive experiments on real-world datasets demonstrate the effectiveness, universality, flexibility and efficiency of our proposed method. Our code repository is available at **https://github.com/Onedean/ST-TTC**.

## 1 Introduction

Spatio-temporal forecasting (STF) aims to predict the future state of dynamic systems from historical spatio-temporal observations and underpins many real-world applications, such as traffic flow forecasting [24], air quality forecasting [51], and energy consumption forecasting [76]. Although spatio-temporal neural networks [32, 33, 61], which couple spatial neural operators with temporal neural operators, have achieved remarkable progress on these tasks, their deployment in practical environments remains fraught with challenges. These observations, typically collected by sensors, are frequently corrupted by noise, outliers (*e.g.*, spikes or dropouts due to hardware failure) [85], and more commonly, non-stationary distribution shifts arising from sensor aging and seasonal patterns [74].

To enhance generalization and performance, prior work has focused primarily on out-of-distribution (OOD) learning for ST data during the training phase: designing architectures that resist perturbations [31, 49, 68, 84], augmenting training data with noise or adversarial examples [3, 46, 97], and introducing specialized loss functions or regularizers [43, 98] to counteract distribution drift. However, these methods share fundamental limitations: they assume that the training data sufficiently captures all future target domain invariance, *a premise that is rarely valid in real-world settings*. Concurrently, an emerging paradigm of continual fine-tuning [10, 11, 35, 38, 69, 70, 71] has become popular in spatio-temporal learning by continuously tuning the model to adapt to dynamic changes. Though promising, it still divides the target domain into multiple periods of training and testing and relies on period-specific training data to optimize model, *thereby failing in data-scarce scenarios*.

---

[*]Y. Liang is the corresponding author.

39th Conference on Neural Information Processing Systems (NeurIPS 2025).

Table 1: Formal comparison of different spatio-temporal learning paradigms for generalization from the perspective of data and learning. $s$ denotes the source domain, $t$ denotes the target domain, $x$ and $y$ denote the samples and labels sampled from $\mathbf{X}$ and $\mathbf{Y}$, respectively. OOD learning expects inputs sampled from any environment $e^* \sim \mathcal{E}$ to be valid, while others are only optimized for the current training or test environment $e$. In particular, continual fine-tuning divides the target domain into multiple stages and optimizes for a specific stage $\tau$ environment $e^\tau$. ✗ means not involved.

| Setting | Example Works | Data Perspective | | Learning Perspective | |
| --- | --- | --- | --- | --- | --- |
| | | Source | Target | Train-Time | Test-Time |
| OOD Learning | STONE [68], CaST [84] | $\langle \mathbf{X}^s, \mathbf{Y}^s \rangle$ | ✗ | $\min\limits_{f_\theta} \max\limits_{e^* \in \mathcal{E}} \mathbb{E}_{(x,y)\sim P(\mathbf{X}^s,\mathbf{Y}^s\mid e^*)}\big[L(f_\theta(x), y)\big]$ | ✗ |
| Continual Fine-Tuning | EAC [10], TrafficStream [11] | ✗ | $\langle \mathbf{X}^t, \mathbf{Y}^t \rangle$ | $\min\limits_{f_{\theta^\tau}} \mathbb{E}_{(x,y)\sim P(\mathbf{X}^t,\mathbf{Y}^t\mid e^\tau)}\big[L(f_{\theta^\tau}(x), y)\big]$ | ✗ |
| Test-Time Training | TTT-ST [9] | $\mathbf{X}^s$ | $\mathbf{X}^t$ | $\min\limits_{f_\theta} \mathbb{E}_{(\tilde{x},x)\sim P(\mathbf{X}^s\mid e)}\big[L(f_\theta(\tilde{x}), x)\big]$ | $\min\limits_{f_\theta} \mathbb{E}_{(\tilde{x},x)\sim P(\mathbf{X}^t\mid e)}\big[L(f_\theta(\tilde{x}), x)\big]$ |
| Online Continual Learning | DOST [72] | ✗ | $\mathbf{X}^t$ | ✗ | $\min\limits_{f_{\theta(\delta)}} \mathbb{E}_{(x,y)\sim P(\mathbf{X}^t\mid e)}\big[L(f_{\theta(\delta)}(x), y)\big]$ |
| Test-Time Computing | `ST-TTC` (Our) | ✗ | $\mathbf{X}^t$ | ✗ | $\min\limits_{g_\theta} \mathbb{E}_{(x,y)\sim P(\mathbf{X}^t\mid e)}\big[L\big(g_\theta(f_\theta(x)), y\big)\big]$ |

$$f_\theta \qquad\qquad x^t$$
$$\downarrow \qquad\qquad \downarrow$$
$$\tilde{x}^t \rightarrow \boxed{\hat{x}^t = f(\tilde{x}^t; \theta)} \rightarrow \hat{x}^t \rightarrow \boxed{Loss(\hat{x}^t, x^t)}$$

(a) Test-Time Training

$$f_{\theta(\delta)} \qquad\qquad y^t$$
$$\downarrow \qquad\qquad \downarrow$$
$$x^t \rightarrow \boxed{\hat{y}^t = f(x^t; \theta(\delta))} \rightarrow \hat{y}^t \rightarrow \boxed{Loss(\hat{y}^t, y^t)}$$

(b) Online Continual Learning

$$f_\theta \qquad\qquad g_\theta \qquad\qquad y^t$$
$$\downarrow \qquad\qquad \downarrow \qquad\qquad \downarrow$$
$$x^t \rightarrow \boxed{\hat{y}^t = f(x^t; \theta)} \rightarrow \hat{y}^t \rightarrow \boxed{\hat{y}_{cal}^t = g(\hat{y}^t; \theta)} \rightarrow \hat{y}_{cal}^t \rightarrow \boxed{Loss(\hat{y}_{cal}^t, y^t)}$$

(c) Test-Time Computing

Red represents the parameters that need to be optimized during the test.

Figure 1: Conceptual visualization comparison of different spatio-temporal learning paradigms under test environment. *(a) Test-Time Training* requires the use of additional pretext tasks in the training and test phases to optimize the self-supervision head or the overall model parameters $f_\theta$. *(b) Online Continual Learning*, by optimizing some internal parameters $f_{\theta(\delta)}$ of the model, requires additional modifications to the internal architecture of the network. Our *(c) Test-Time Computing* method only requires a lightweight calibrator $g_\theta$, which is a seamless and lightweight plug-and-play module.

Recently, leveraging test-time information has attracted widespread attention for its ability to significantly improve language model performance on complex reasoning tasks [1, 64]. In computer vision, this concept has already been extensively developed: test-time training (TTT) was first introduced by [66], which defines an auxiliary self-supervised task applied to both training and test samples to better balance bias and variance [20]. A similar idea was adapted to spatio-temporal forecasting in TTT-ST [9]. Unlike language and vision settings—where obtaining ground-truth labels for test samples at inference time is nearly impossible—STF benefits from label autocorrelation [14]: each observation strongly depends on its predecessor, and training instances are constructed from sliding windows, which provide access to historical samples and their true labels. Moreover, this property makes STF also require timeliness [60], that is, the additional computing time during inference must be less than the window-stride interval. A recent STF method, DOST [72], explores online continual learning, which initially explored this direction. It uses historical test sample labels to dynamically adapt the modified model architecture. *Though promising, these approaches typically involve complex self-supervised tasks or structural adaptations and still fall short of the timeliness demands of STF.*

To address this gap, we propose Test-Time Computing of Spatio-Temporal Forecasting (`ST-TTC`), an attractive complementary paradigm. `ST-TTC` achieves learning with calibration by iteratively leveraging available test information during inference, enabling seamless integration with diverse models. This adapts the model to evolving spatio-temporal patterns, thereby calibrating predictions. Our principal insight is that performance degradation during test time is primarily driven by non-stationary distributional shifts stemming from progressive periodic biases. Therefore, we propose a spectral domain calibrator. This involves appending a lightweight module, operating in the frequency

domain, subsequent to the backbone network. This module calibrates biases by learning minor, node-specific amplitude and phase correction factors. Furthermore, a flash gradient updating mechanism with a streaming memory queue, ensures universal, rapid, and resource-efficient test-time computing. Table 1 provides a formal comparison of our method against existing learning paradigms, and Figure 1 offers a conceptual visualization of learning with test domain. In summary, our contributions are:

- We propose a novel test-time computing paradigm of spatio-temporal forecasting, termed `ST-TTC` .

- We systematically explore the goals and means of achieving this paradigm. Concretely, we introduce a spectral domain calibrator with phase-amplitude modulation to mitigate periodic shift and present a flash updating mechanism with a streaming memory queue for efficient test-time computation.

- Experimental results on real-world spatio-temporal datasets in different fields, scenarios, and learning paradigms demonstrate the effectiveness and universality of `ST-TTC` .

## 2 Related Work

**Spatio-Temporal Forecasting.** Spatio-temporal sequences can be regarded as spatially extended multivariate time series. Although one can trivially apply multivariate forecasting methods [5, 7, 52, 96] independently at each location, such decoupling of spatial and temporal dependencies invariably yields suboptimal results [61]. Classical spatio-temporal forecasting method instead relies on shallow models or spatio-temporal kernels, including feature-based methods [53, 102], state space models [2, 13, 56], and Gaussian process models [18, 58]. Unfortunately, the overall nonlinearity of these models is limited, and the high complexity of computation and storage further hinders the availability of massive training instances [63]. In recent years, spatio-temporal neural networks [32, 33, 36] have been widely adopted to learn the complex dynamics of such systems. Early work concentrated on devising neural operators to extract spatial or temporal correlation [17, 47, 62, 83, 90] and on designing fusion architectures to integrate them [12, 16, 23, 40, 54]. More recent efforts have explored domain-invariant representation learning [49, 103, 104] and continual model adaptation [10, 11, 89] to better accommodate unseen environmental shifts. *However, these methods still depend exclusively on offline training data and thus cannot deliver truly timely and effective adaptation in real settings.*

**Test-Time Computing.** Test-time computation is inspired by the human cognition [34], in which additional computational effort is allocated during inference to improve task performance. This insight has recently driven considerable interest in the nature language process community, fueled by the success of reasoning-augmented language models (*e.g.*, o1 [29] and r1 [21]) that activate and adapt internal computations at test time via supervised fine-tuning or reinforcement learning (RL) [99]. While the generalization properties of RL-based adaptation remain debated [95], the notion of supervised learning on unlabeled test data dates back to "transductive learning" [19] in the 1990s and has demonstrated empirical benefits [6, 67]. In the computer vision domain, this idea was formalized as Test-Time Training [66], which attaches an auxiliary self-supervised head to enable online adaptation to each test instance—a paradigm subsequently generalized as test-time adaptation [30, 44, 73, 77]. However, spatio-temporal forecasting has seen limited exploration of such techniques. TTT-ST [9] applies TTT-style auxiliary objectives during training and continues to update at inference, and DOST [72] further incorporates dynamic learning mechanisms within modified model architectures for test-time updates. In addition, some methods [22, 100] are conceptually close to ours, such as CompFormer [100], which proposes a test-time compensated representation learning framework, but still requires access to additional training data. *Notably, we formalize the test-time computing of spatio-temporal forecasting, and propose a unified learning-with-calibration framework that is general, lightweight, efficient, and effective for STF at test-time.*

For more related work, we provide a more detailed introduction in Appendix A.

## 3 Preliminaries

**Problem Definition.** Let $x \in \mathbb{R}^{T \times C}$ denote the multivariate time series recorded at each location sensor, capturing the dynamic observations of $C$ measured features in $T$ consecutive time steps. Stacking these sequences for all $N$ locations yields the spatio-temporal tensor $X \in \mathbb{R}^{N \times T \times C}$. Given historical observations $X^h \in \mathbb{R}^{N \times T^h \times C}$ (and an optional spatial correlation graph $\mathcal{G}$ representing the spatial relationships of $N$ locations), spatio-temporal forecasting aims to learn a mapping $f_\theta$ :

$(X^h, \mathcal{G}) \longmapsto X^f \in \mathbb{R}^{N \times T^f \times C}$, where $X^f$ is the signal for the next $T^f$ time steps. In practice, according to [40, 61], the feature to be predicted is usually only the target variable.

**Scenario Definition.** In deep learning systems, batch-based testing is typically employed to exploit parallelism. In real-world deployment, however, predictions must be produced for each incoming time-step sample—*i.e.*, with batch size $B$ set to 1. At time index $t$, once the new sliding-window input $X_t \in \mathbb{R}^{N \times T^h \times C}$ arrives, the true labels for all test samples before time index $t - T^h - T^f + 1$ become available. Thus, test-time computing of spatio-temporal forecasting can leverage this accumulated historical information to enhance the accuracy of the current prediction, while ensuring that any additional computation latency remains below a threshold defined by the sliding-window stride.

# 4 Methodology

Our test-time computing framework of spatio-temporal forecasting (`ST-TTC`) integrates two synergistic components: 1) a spectral domain calibrator with phase-amplitude modulation; and 2) a flash gradient update mechanism with streaming memory queue. In this section, we introduce these two key components, respectively, from the perspective of what is computed and how it is computed.

## 4.1 What to Compute? Spectral Domain Calibrator with Phase-Amplitude Modulation

**Motivation.** Spatio-temporal data, such as traffic flow and air quality, often exhibit periodic patterns (*e.g.*, daily or weekly cycles). However, in real-world deployments, these patterns are not stationary; they are dynamically influenced by various internal and external factors [75]. Such influences lead to non-stationarities manifesting as fluctuations in amplitude (*e.g.*, increased or decreased traffic peaks due to seasonal changes) or phase shifts (*e.g.*, peak hours are advanced or delayed due to traffic congestion). Pre-training models typically fit fixed periodic patterns during training, which makes them vulnerable to performance degradation under such persistent dynamic changes during inference [74]. Therefore, we argue that *the goal of test-time computation is: how to design an effective calibrator that can efficiently capture such gradual systematic bias from the pattern to correct the prediction errors caused by non-stationarity, while avoiding overfitting to random noise?*

**Key Challenges.** While correction in the time domain is possible [22, 100], it often requires extensive parameterization, leading to increased model complexity and limited ability to capture evolving periodic structures. Moreover, the coupled structural and branching modules [9, 72] are prone to overfitting the random noise in the spatio-temporal evolution. To address this, we propose calibration in the spectral domain, where periodic variations are more transparently expressed as changes in the amplitude and phase of specific frequency components. Spectral correction offers a potentially more direct and robust solution. However, this introduces two main challenges: ❶ the degree of non-stationarity varies across spatial nodes; and ❷ full-spectrum parameterization is computationally expensive. *The core problem thus becomes how to design a lightweight, spatial-aware calibrator.*

**Implementation Details.** To this end, we formally introduce the *spectral domain calibrator (SD-Calibrator)*, which is a lightweight plug-and-play module that performs spectral domain calibration on the time domain prediction results of the pre-trained model, aiming to achieve efficient test-time computation for spatio-temporal forecasting. Specifically, it can be divided into three steps:

- *Spatial-aware Decomposition.* To ensure spatial awareness, we apply a real-to-complex fast Fourier transform (rFFT) along the time dimension of the backbone model's prediction $\hat{y} \in \mathbb{R}^{B \times N \times T}$, separately for each spatial node. This yields the frequency spectrum: $Y_f = \text{rFFT}(\hat{y}) \in \mathbb{C}^{B \times N \times M}$, where $M = \frac{T}{2} + 1$ is the number of unique frequency bins for real-valued signals. Then, we decompose $Y_f$ into its amplitude $A = |Y_f| \in \mathbb{R}^{B \times N \times M}$ and phase $P = \angle Y_f \in \mathbb{R}^{B \times N \times M}$.

- *Group-wise Modulation.* To ensure lightweight and balanced spectrum expression, we divide the $M$ frequency bins into $G$ contiguous groups of size $\lfloor M/G \rfloor$, and learn per-group, per-node amplitude and phase offsets $\lambda^\alpha \in \mathbb{R}^{G \times N \times 1}, \lambda^\phi \in \mathbb{R}^{G \times N \times 1}$ (Note: Both $\lambda^\alpha$ and $\lambda^\phi$ are initialized to 0 to avoid incorrect calibration of predictions before learning). For each group $g \in \{1, \ldots, G\}$, we apply $A'_g = A_g \odot (1 + \lambda^\alpha_g), P'_g = P_g + \lambda^\phi_g$, and reconstruct the spectrum as $Y'_f = \bigcup_{g=1}^{G} A'_g \odot e^{(j\, P'_g)}$.

- *Inverse Transform.* Finally, the calibrated time-domain signal is obtained by Inverser rFFT $\hat{y}_{cal} = \text{irFFT}(Y_f) \in \mathbb{R}^{B \times N \times T}$, along the frequency dimension.

For clarity, we provide a Algorithm workflow 1 and Pytorch-Style Pseudocode 2 in Appendix C.1.

**Complexity Analysis.** The full-spectrum parameterization learns independent amplitude and phase offsets for each of the $M = T/2 + 1$ frequency bins and $N$ nodes, totaling $2NM$ parameters. In contrast, our $G$-group design learns only $2NG$ parameters. Since $G$ is a constant and $M$ grows linearly with $T$, $G \ll M$ is usually the case. For large-scale long-term scenario, this significantly reduces memory footprint and gradient update cost while retaining interpretable per-band calibration.

**Theoretical Analysis.** We also provide a theoretical approximate bound on the output perturbation induced by the *SD-Calibrator*, ensuring controlled deviation from the original prediction to prevent overfitting (Please refer to Theorem 1 and the proof in Appendix B).

## 4.2 How to Compute? Flash Gradient Update with Streaming Memory Queue

**Motivation.** The *SD-Calibrator* provides an effective mechanism for output correction. To accommodate the dynamic nature of spatio-temporal data, its parameters $(\lambda^\alpha, \lambda^\phi)$ must be continuously updated during inference. Fortunately, as we discussed above, due to the streaming nature of spatio-temporal data, unlike Visual and textual tasks, we have access to the true labels of historical samples. However, simply accumulating all historical data for updates is not feasible due to the increasing computational load and memory usage. Therefore, we argue that *the key to test-time computation is: How to design an efficient data selection and learning mechanism that leverages appropriate historical information to tuning the SD-Calibrator without incurring a lot of computational overhead?*

**Key Challenges.** Although retrieving similar sequences from historical training databases can partially compensate for prediction errors [100], this assumption is unrealistic, as only test-time information is available in our scenario. Moreover, selectively storing historical test samples via memory bank primarily serves to mitigate catastrophic forgetting in the backbone model [72], which misaligns with the learning objective of our *SD-calibrator*. To address this, we propose freezing the backbone and updating only the calibrator using recent test samples for efficient test-time computing. However, this strategy introduces two critical challenges: ❶ recent studies [37] have shown that real-time updates may cause information leakage; and ❷ excessive updates can lead to overfitting of the calibration parameters and increased computational burden. *The core problem thus becomes how to design a efficient calibration parameter learning mechanism without information leakage.*

**Implementation Details.** To address these challenges, we introduce the *flash gradient update* strategy coupled with a *streaming memory queue*. The process is as follows:

- *Streaming Memory Queue.* We maintain a first-in, first-out (FIFO) queue, denoted as $\mathcal{Q}$, with a maximum size equal to the prediction horizon $T^f$. For each incoming test instance $t$, after making a prediction, we store the input-label pair $(X_t, Y_t)$ into $\mathcal{Q}$ (Here is for engineering convenience. In real deployment, data points can be merged at each step to form the true label). Once $\mathcal{Q}$ is full, for every new test sample $(X_n, Y_n)$ added, the oldest sample pair $(X_o, Y_o)$ is dequeued. This dequeued sample $(X_o, Y_o)$ is then used for the gradient update, thus avoiding the information leakage.

- *Flash Gradient Update.* Once we have $(X_o, Y_o)$, we first obtain the backbone model's prediction for the historical input: $\hat{Y}_o^b = f_\theta(X_o)$ (note: the backbone model weights $f_\theta$ are frozen). Then, the *SD-Calibrator* $g_\theta$ processes this prediction: $\hat{Y}_o^{cal} = g_\theta(\hat{Y}_o^b)$. The loss function between the calibrated prediction $\hat{Y}_o^{cal}$ and the true historical label $Y_o$ is calculated, and only a single gradient descent step is performed to update the parameters of the *SD-Calibrator*: $\lambda \leftarrow \lambda - \eta \nabla_\lambda L$. For the next input sample $X_t$, the updated *SD-Calibrator* is used for prediction. Using this single-sample single-step gradient descent strategy, we achieve lightning-fast parameter updates.

For clarity, we provide a Algorithm workflow 3 and Pytorch-Style Pseudocode 4 in Appendix C.2.

**Complexity Analysis.** The primary focus here is the time complexity. The *Streaming Memory Queue* itself has an $\mathcal{O}(1)$ time complexity for enqueue and dequeue operations. The *Lightning Gradient Update* is performed only once for each incoming test sample. Each update involves: 1). Forward propagation of the backbone and calibrator (dominated by the computational cost $\mathcal{O}(NT \log T)$ of rFFT and irFFT) 2). Backward propagation of the calibrator (dominated by parameter cost $\mathcal{O}(NG)$).

**Theoretical Analysis.** We also show that this single update step leads to a controlled adjustment, ensuring that the calibrator makes progress on the newest sample it's trained on, without causing erratic behavior, under standard assumptions. (Please refer to Proposition 2 in Appendix B).

# 5 Experiments

In this section,we conduct extensive experiments to answer the following research questions (RQs):

- **RQ1:** Can ST-TTC have a consistent improvement on various types of models and datasets? Can ST-TTC outperform previous learning methods that leverage test data? *(Effectiveness)*

- **RQ2:** Can ST-TTC effective in various real-world scenarios, including few-shot learning, long-term forecasting, and large-scale forecasting? *(Universality)*

- **RQ3:** Can ST-TTC further enhance the performance of existing learning paradigms that utilize training data, such as OOD Learning and continual learning? *(Flexibility)*

- **RQ4:** How does ST-TTC work? Which components or strategies are crucial? Are these components or strategies sensitive to parameters or design? *(Mechanism & Robustness)*

- **RQ5:** What is the time and parameter cost of ST-TTC during test-time computation, and how does it compare to other advanced methods? *(Efficiency & Lightweight)*

## 5.1 Experimental Setup

**Datasets.** We employ publicly available benchmark datasets widely used in the literature to cover typical spatio-temporal forecasting scenarios in the traffic domain (*PEMS-03*, *PEMS-04*, *PEMS-07*, *PEMS-08* [65]), the meteorological domain (*KnowAir* [79]), and the energy domain (*UrbanEV* [39]). In addition, we also leverage the traffic-speed benchmark *METR-LA* [40], the large-scale spatio-temporal benchmark *LargeST* [48], and dynamic-stream benchmarks (*Energy-Stream*, *Air-Stream*, *PEMS-Stream* [10]) to assess our methods across varied settings and learning paradigms. Unless otherwise specified, all datasets are chronologically split into training, validation and test sets in a 6 : 2 : 2 ratio. For more detailed description of each dataset, please see the Appendix D.1.

**Baseline.** For the default evaluation, we cover various widely used spatio-temporal backbones, which can be divided into three categories: (1) Transformer-based: *STAEformer* [47] and *STTN* [86]; (2) Graph-based: *GWNet* [83] and *STGCN* [90]; (3) MLP-based: *STID* [62] and *ST-Norm* [15]. For the baselines that leverage test information, we cover three types: (1) popular test-time adaptation methods in vision: *TTT-MAE* [20] and *TENT* [73]; (2) Online time series forecasting methods: *OnlineTCN* [105], *FSNet* [57] and *OneNet* [81]; (3) Comparable online spatio-temporal forecasting methods: *CompFormer* [100] and *DOST* [72]. For the baselines on large-scale benchmarks, we use the efficient *PatchSTG* [17] as the backbone. For the baseline of OOD learning scenarios, we use the advanced *STONE* [68] as the default method. For the continual learning scenario, we use *EAC* [10] and *STKEC* [70] as the default methods. We follow the default parameter settings of the models for all scenarios according to the corresponding literature. For details of each method, see Appendix D.2.

**Protocol.** Following prior benchmarks [61], we employ a 12-to-12 forecasting protocol—using the previous 12 time steps to predict the next 12 steps and their mean—evaluated with mean absolute error (MAE), root mean square error (RMSE), and mean absolute percentage error (MAPE). For simplicity, all experiments share the same hyperparameters of our ST-TTC : the calibration module learning rate $lr$ is set to 1e-4, the memory-queue sample count $n$ used for updating is 1, and the number of groups $m$ to 4. To ensure fairness, each experiment is repeated five times, with results reported as mean ± standard deviation (denoted in gray ±). More protocol details, see Appendix D.3.

## 5.2 Effectiveness Study (RQ1)

**Consistent Effectiveness.** Table 2 presents the results of our method for 12-step future prediction across six models on six public datasets. The ✗ column denotes the results of standard testing, while the ✓ column indicates results obtained with our proposed ST-TTC approach. The best results in the ✗ and ✓ columns are highlighted in bold **blue** and **pink** fonts, respectively. We also compute the relative improvement, denoted by the $\Delta$ column. Based on these results, we make the following observations: ❶ The application of our test-time computation method, ST-TTC , consistently yields performance gains across various backbone architectures and dataset combinations. ❷ From a model-centric perspective, our approach can further enhance the **performance** of even the **top-performing** methods across different metrics and datasets. ❸ From a data-centric perspective, *UrbanEV* shows more significant relative improvement, likely due to its more pronounced distribution shift.

Table 2: Performance comparison of different models w/ and w/o `ST-TTC` on common benchmarks.

| Models | | | Transformer-based | | | | | | Graph-based | | | | | | MLP-based | | | | | |
|---|---|---|---|---|---|---|---|---|---|---|---|---|---|---|---|---|---|---|---|---|
| | | STAEformer [47] | | | STTN [86] | | | GWNet [83] | | | STGCN [90] | | | STID [62] | | | ST-Norm [15] | | |
| w/ `ST-TTC` | | ✗ | ✓ | Δ(%) | ✗ | ✓ | Δ(%) | ✗ | ✓ | Δ(%) | ✗ | ✓ | Δ(%) | ✗ | ✓ | Δ(%) | ✗ | ✓ | Δ(%) |
| PEMS-03 | MAE | 17.00±0.16 | 16.75±0.14 | ↓1.47 | 18.12±0.53 | 17.88±0.50 | ↓1.32 | 16.73±0.41 | 16.42±0.19 | ↓1.85 | 18.41±0.35 | 18.08±0.35 | ↓1.79 | 17.48±0.02 | 17.29±0.01 | ↓1.09 | 17.27±0.13 | 17.03±0.12 | ↓1.39 |
| | RMSE | 29.98±0.59 | 29.48±0.58 | ↓1.67 | 31.02±1.55 | 30.48±1.37 | ↓1.74 | 28.48±0.48 | 27.90±0.13 | ↓2.04 | 31.74±0.94 | 31.10±0.86 | ↓2.02 | 29.10±0.22 | 28.74±0.21 | ↓1.24 | 29.28±0.20 | 28.71±0.14 | ↓1.95 |
| | MAPE(%) | 15.82±0.22 | 15.82±0.20 | ↓0.00 | 18.43±1.19 | 18.01±1.06 | ↓2.28 | 16.70±0.60 | 16.49±0.30 | ↓1.26 | 18.90±0.78 | 18.68±0.44 | ↓1.16 | 17.50±0.12 | 17.39±0.01 | ↓0.63 | 17.20±0.82 | 16.92±0.52 | ↓1.63 |
| PEMS-04 | MAE | 19.48±0.05 | 19.33±0.06 | ↓0.77 | 20.63±0.03 | 20.48±0.03 | ↓0.73 | 20.57±0.37 | 20.49±0.39 | ↓0.39 | 20.70±0.14 | 20.57±0.12 | ↓0.63 | 19.97±0.07 | 19.86±0.06 | ↓0.55 | 20.22±0.10 | 20.08±0.09 | ↓0.69 |
| | RMSE | 32.58±0.39 | 32.31±0.34 | ↓0.83 | 33.14±0.16 | 32.82±0.10 | ↓0.97 | 32.64±0.35 | 32.49±0.38 | ↓0.46 | 33.11±0.22 | 32.80±0.20 | ↓0.94 | 32.62±0.08 | 32.49±0.08 | ↓0.40 | 33.15±0.27 | 32.73±0.23 | ↓1.27 |
| | MAPE(%) | 12.42±0.01 | 12.37±0.09 | ↓0.40 | 14.74±0.16 | 14.53±0.41 | ↓1.42 | 14.44±0.47 | 14.43±0.32 | ↓0.07 | 14.15±0.16 | 14.04±0.15 | ↓0.78 | 12.78±0.17 | 12.72±0.06 | ↓0.47 | 13.72±0.13 | 13.68±0.24 | ↓0.29 |
| PEMS-07 | MAE | 21.67±0.16 | 21.41±0.13 | ↓1.20 | 23.30±0.81 | 23.08±0.75 | ↓0.94 | 22.62±0.27 | 22.46±0.27 | ↓0.71 | 24.26±0.23 | 23.83±0.18 | ↓1.77 | 21.72±0.05 | 21.55±0.05 | ↓0.78 | 22.69±0.15 | 22.50±0.14 | ↓0.84 |
| | RMSE | 37.48±0.52 | 37.03±0.48 | ↓1.20 | 37.55±0.91 | 37.24±0.81 | ↓0.83 | 36.87±0.23 | 36.62±0.26 | ↓0.68 | 39.31±0.26 | 38.63±0.20 | ↓1.73 | 36.24±0.06 | 36.00±0.06 | ↓0.66 | 38.14±0.44 | 37.77±0.41 | ↓0.97 |
| | MAPE(%) | 8.92±0.05 | 8.87±0.06 | ↓0.56 | 10.08±0.24 | 9.98±0.24 | ↓0.99 | 9.79±0.16 | 9.75±0.11 | ↓0.41 | 10.57±0.18 | 10.38±0.07 | ↓1.80 | 9.05±0.03 | 9.00±0.03 | ↓0.55 | 9.94±0.56 | 9.86±0.33 | ↓0.80 |
| PEMS-08 | MAE | 14.84±0.09 | 14.73±0.08 | ↓0.74 | 17.19±0.17 | 17.07±0.16 | ↓0.70 | 16.37±0.21 | 16.28±0.20 | ↓0.55 | 17.33±0.19 | 17.17±0.21 | ↓0.92 | 15.61±0.02 | 15.52±0.02 | ↓0.58 | 16.69±0.06 | 16.56±0.04 | ↓0.78 |
| | RMSE | 25.61±0.17 | 25.49±0.16 | ↓0.47 | 27.07±0.30 | 26.93±0.29 | ↓0.52 | 26.14±0.23 | 26.05±0.21 | ↓0.34 | 27.49±0.21 | 27.31±0.22 | ↓0.65 | 25.70±0.05 | 25.60±0.05 | ↓0.39 | 26.94±0.10 | 26.80±0.10 | ↓0.52 |
| | MAPE(%) | 9.37±0.04 | 9.32±0.06 | ↓0.53 | 11.27±0.26 | 11.20±0.28 | ↓0.62 | 10.95±0.34 | 10.79±0.14 | ↓1.46 | 11.63±0.16 | 11.53±0.30 | ↓0.86 | 9.82±0.06 | 9.76±0.06 | ↓0.61 | 11.46±0.89 | 11.19±0.38 | ↓2.36 |
| KnowAir | MAE | 17.13±0.19 | 17.06±0.18 | ↓0.41 | 17.14±0.16 | 17.06±0.16 | ↓0.47 | 17.03±0.07 | 16.94±0.07 | ↓0.53 | 17.03±0.06 | 16.96±0.06 | ↓0.41 | 18.07±0.11 | 17.98±0.10 | ↓0.50 | 17.07±0.01 | 17.01±0.02 | ↓0.35 |
| | RMSE | 26.13±0.20 | 26.06±0.19 | ↓0.27 | 26.18±0.16 | 26.13±0.16 | ↓0.19 | 26.12±0.15 | 26.05±0.14 | ↓0.27 | 26.14±0.17 | 26.07±0.17 | ↓0.27 | 27.23±0.02 | 27.17±0.02 | ↓0.22 | 26.45±0.07 | 26.39±0.04 | ↓0.23 |
| | MAPE(%) | 62.14±1.62 | 61.66±1.53 | ↓0.77 | 64.12±1.24 | 63.15±1.40 | ↓1.51 | 64.51±0.20 | 63.62±0.28 | ↓1.38 | 63.12±0.97 | 62.70±0.84 | ↓0.67 | 70.00±1.44 | 69.07±1.35 | ↓1.33 | 60.76±0.56 | 60.50±0.63 | ↓0.43 |
| UrbanEV | MAE | 2.87±0.02 | 2.85±0.02 | ↓0.70 | 3.04±0.06 | 2.99±0.06 | ↓1.64 | 2.89±0.03 | 2.85±0.03 | ↓1.38 | 3.29±0.10 | 3.23±0.10 | ↓1.82 | 2.83±0.01 | 2.79±0.01 | ↓1.41 | 3.09±0.02 | 3.04±0.02 | ↓1.62 |
| | RMSE | 5.00±0.01 | 4.98±0.02 | ↓0.40 | 5.09±0.07 | 5.03±0.07 | ↓1.18 | 4.87±0.07 | 4.81±0.06 | ↓1.23 | 5.63±0.23 | 5.52±0.22 | ↓1.95 | 4.74±0.02 | 4.67±0.02 | ↓1.48 | 5.31±0.03 | 5.22±0.02 | ↓1.69 |
| | MAPE(%) | 27.14±0.46 | 26.75±0.58 | ↓1.44 | 28.75±0.08 | 28.22±0.29 | ↓1.84 | 29.10±0.33 | 28.47±0.26 | ↓2.16 | 31.67±0.97 | 31.45±1.01 | ↓0.70 | 27.60±0.62 | 27.15±0.43 | ↓1.63 | 29.53±0.57 | 29.26±0.57 | ↓0.91 |

**Competitive Effectiveness.** We further compare our method against various advanced approaches that can leverage test-time information. Since the official source code of *CompFormer* and *DOST* is not available and uses additional data information, it leads to an unfair comparison. Nevertheless, we still include all reported *METR-LA* benchmark values (indicated with **\***) using a unified *GWNet* backbone, categorized into regular and online settings as presented in Table 3, based on their respective papers. Additionally, we implemented the popular *TTT-MAE* method as a surrogate for the unavailable *TTT-ST* method. Our observations are as follows: ❶ For the regular setting, our method achieves competitive results with more stable standard deviations. While other methods like *CompFormer* demonstrate similar performance, they often utilize more training information and computational resources. ❷ In the online setting, our method significantly outperforms existing approaches without requiring more complex model architecture modifications.

Table 3: Performance comparison of the advanced method with `ST-TTC` on *METR-LA* benchmark. Marker $^\dagger$ indicates the results are statistically significant (t-test with p-value < 0.01).

| Method | MAE | RMSE | w/o Training Set | w/o Modifying Backbone |
|---|---|---|---|---|
| *The training / validation / test set split used below is 70% / 10% / 20%.* | | | | |
| ***TTT-MAE*** [20] | 3.47±0.03 | 7.43±0.05 | ✗ | ✓ |
| ***TENT*** [73] | 4.84±0.08 | 8.53±0.10 | ✓ | ✓ |
| ***CompFormer**** [100] | 3.46±0.02 | 7.19±0.08 | ✗ | ✓ |
| `ST-TTC` | 3.46±0.01$^\dagger$ | 7.21±0.01 | ✓ | ✓ |
| *The training / validation / test set split used below is 20% / 5% / 75%.* | | | | |
| ***OnlineTCN**** [105] | 4.78±0.03 | 8.70±0.04 | ✓ | ✗ |
| ***FSNet**** [57] | 5.79±0.24 | 11.06±0.24 | ✓ | ✗ |
| ***OneNet**** [81] | 4.94±0.03 | 8.80±0.06 | ✓ | ✗ |
| ***DOST**** [72] | 4.38±0.02 | 8.26±0.03 | ✓ | ✗ |
| `ST-TTC` | 3.77±0.07$^\dagger$ | 7.75±0.13$^\dagger$ | ✓ | ✓ |

## 5.3 Universality Study (RQ2)

To demonstrate the universality of `ST-TTC` across diverse real-world scenarios, we explore various forecasting scenarios in the literature, including few-shot [94], long-term [59], and large-scale [25].

**Few-Shot Scenario.** To simulate limited training data, we retrained models using only the first 10% of existing training sets to investigate a more common and challenging few-shot scenario. Figure 2 shows the relative performance gains with `ST-TTC` (For full results, please refer Table 6 in the Appendix). We observe: ❶ `ST-TTC` provides more significant improvements in the few-shot setting compared to the full-shot case in Table 2, with about half exceeding 2%. ❷ *KnowAir* shows the largest gain compared to other datasets, likely because its four-year long period leads to a substantial test distribution shift in the few-shot scenario, where our method adapts well.

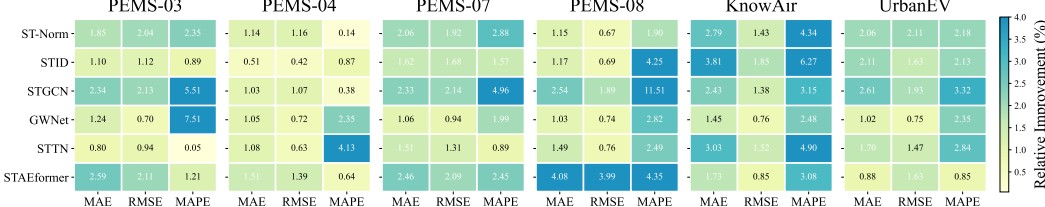

Figure 2: Relative improvements of different models w/ `ST-TTC` in the few-shot setting.

**Long-Term Scenario.** In real-world scenarios, long-term forecasting helps to further plan future decisions. We predicted 24 future steps from 24 past steps to explore more complex temporal changes. As shown in Figure 3, we present the relative performance improvement of the advanced *STID* model

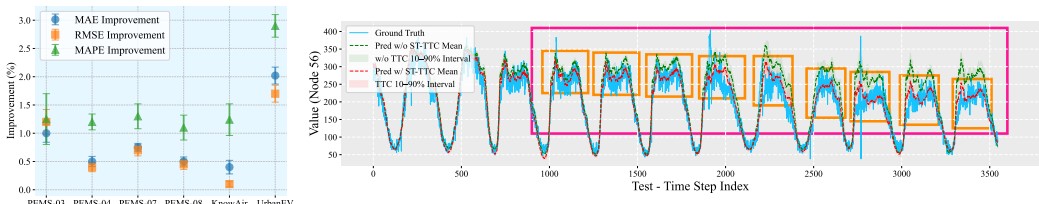

Figure 3: Left: relative improvement of long-term setting. Right: visualization study of *PEMS-08*.

with our `ST-TTC` method, and give a test set prediction visualization case on the *PEMS-08* dataset (see Figure 9 in the Appendix for more examples). Our observations include: ❶ `ST-TTC` consistently improves long-term forecasting, even more than short-term (Table 2), likely due to more learnable information in longer windows. ❷ As the pink and orange box shows, our method learns test-time history, capturing both the global traffic decline and local fluctuations, leading to effective calibration.

**Large-Scale Scenario.** Beyond current regional datasets, state or national-level spatio-temporal forecasting can involve tens of thousands of stations and longer time frames. We explore large-scale scenarios using the popular *LargeST* benchmark (comprising *SD*, *GBA*, *GLA*, and *CA* subsets). Figure 4 illustrates the 12-step prediction performance gains of the state-of-the-art efficient spatio-temporal model *PatchSTG* [17] with our `ST-TTC` , along with a comparison of inference time complexity (For full results, see Table 7 in Appendix). We observe: ❶ Our `ST-TTC` consistently yields further performance improvements across all datasets, even surpassing the improvement of the second-best baseline over the PatchSTG on some datasets. ❷ The additional inference time is at most 3.82 minutes, which is a clear advantage for the achieved performance gains compared to the training time cost of up to 14 hours.

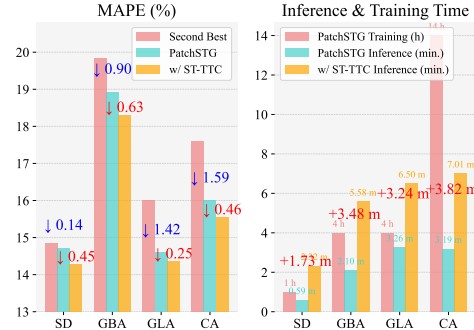

Figure 4: Performance on *LargeST*.

## 5.4 Flexibility Study (RQ3)

To illustrate the flexibility of `ST-TTC` in accommodating existing learning paradigms, we explore its integration with two training data-leveraging paradigms: OOD learning and Continual Learning.

**OOD Learning Setting.** Following prior work [68], we use the *SD* dataset to simulate spatio-temporal shift. For the temporal dimension, we use 1-8/2019, 9-10/2019, and 11-12/2020 for training, validation, and testing, respectively. For the spatial dimension, we randomly mask 10% of nodes in the test set and consider three proportions of new nodes (10% / 15% / 20%) relative to the training node to mimic varying degrees of shift. In Figure 5, we present the 12-step average prediction performance gains of the advanced OOD learning model *STONE* with our `ST-TTC` , evaluated on all nodes and new nodes to demonstrate generalizability and scalability (Full results in Table 8).

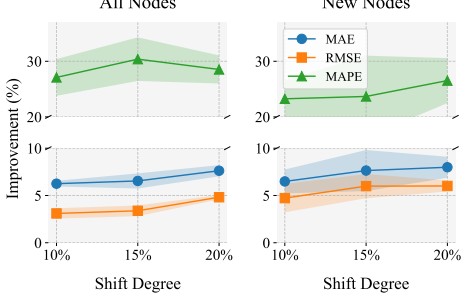

Figure 5: Relative improvement using our `ST-TTC` in the OOD learning setting.

We observe that: ❶ The *STONE* model with `ST-TTC` consistently achieves performance benefits, significantly outperforming all previous settings, indicating that existing OOD models are still insufficient for true OOD generalization, while our method is highly effective. ❷ For both all and new nodes, our improvements become more pronounced as the shift increases, further demonstrating our effectiveness in handling both generalizability and scalability in challenging scenarios.

**Continual Learning Setting.** Following prior work [10], we used multi-period streaming spatio-temporal data to examine our `ST-TTC` 's integration with continual learning method. Table 4 shows the improved 12-step forecasting of advanced continual learning models *EAC* and *STKEC* with our `ST-TTC` . We observed: ❶ Consistent performance gains for both models across all datasets; *STKEC* with `ST-TTC` even achieved comparable performance to best model *EAC*. (2) *Energy-*

Table 4: Performance comparison in continual learning setting.

| Methods | w/ ST-TTC | Air-Stream | | | PEMS-Stream | | | Energy-Stream | | |
|---|---|---|---|---|---|---|---|---|---|---|
| | | MAE | RMSE | MAPE(%) | MAE | RMSE | MAPE(%) | MAE | RMSE | MAPE(%) |
| EAC | ✗ | 24.15±0.14 | 38.22±0.31 | 31.79±0.05 | 14.92±0.11 | 24.17±0.17 | 20.82±0.16 | 5.15±0.10 | 5.46±0.09 | 50.55±2.60 |
| | ✓ | 23.54±0.15 | 37.51±0.27 | 31.44±0.10 | 14.71±0.07 | 23.87±0.12 | 20.53±0.03 | 3.47±0.01 | 3.94±0.00 | 39.66±0.41 |
| | Δ | ↓2.5% | ↓1.9% | ↓1.1% | ↓1.4% | ↓1.2% | ↓1.4% | ↓32.6% | ↓27.8% | ↓21.6% |
| STKEC | ✗ | 25.44±1.05 | 40.11±1.13 | 33.30±1.64 | 16.25±0.04 | 26.73±0.07 | 22.33±0.16 | 5.41±0.15 | 5.72±0.10 | 52.40±1.10 |
| | ✓ | 24.26±0.03 | 39.02±0.04 | 31.73±0.04 | 16.05±0.06 | 26.39±0.11 | 21.88±0.06 | 3.83±0.09 | 4.28±0.08 | 43.22±0.90 |
| | Δ | ↓4.6% | ↓2.7% | ↓4.7% | ↓1.2% | ↓1.3% | ↓2.0% | ↓29.2% | ↓25.2% | ↓17.5% |

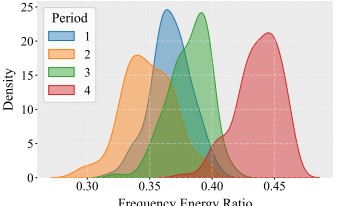

Figure 6: *Energy-Stream*'s Shift.

*Stream* achieves significant improvement over other datasets, as the ST-TTC effectively learns and calibrates temporal changes, as shown by the frequency analysis (drastic shift changes) in Figure 6.

## 5.5 Mechanism & Robustness Study (RQ4)

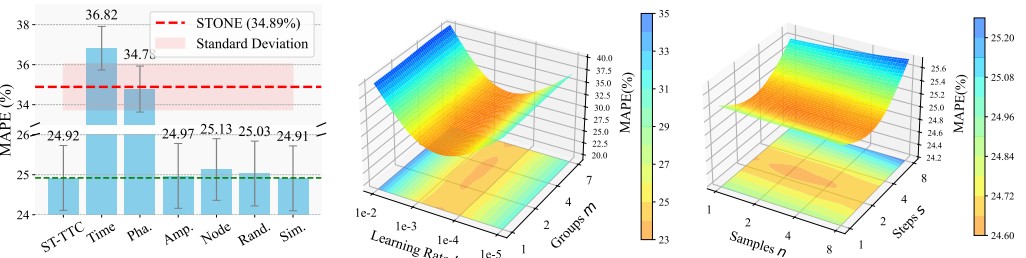

Figure 7: Left: Strategy comparison. Middle: Effect of $lr$ $w.r.t$ $m$ . Right: Effect of $n$ $w.r.t$ $s$.

We follow the OOD setup (challenging setting with 20% new nodes) to evaluate our ST-TTC .

**Strategy Study.** We compare different strategies: 1) simple nonlinear time domain calibration (*Time*), 2) learning only phase or amplitude modulation factors (*Pha. / Amp.*), 3) node-share modeling (*Node*), and (4) random selection or retrieval of the most similar samples (*Rand. / Sim.*). As shown in Figure 7 left, we observe: ❶ Frequency-domain calibration significantly outperforms time-domain calibration, with amplitude modulation being the primary contributor; ❷ Sharing nodes leads to performance degradation due to spatial heterogeneity in spatio-temporal data; ❸ Random sample selection reduces performance, and retrieving similar samples offers negligible gains while incurring higher computational cost. Our proposed update strategy is already near-optimal.

**Parameter study.** We analyze the sensitivity of two parameter groups. As shown in the middle and right of Figure 7: ❶ Higher learning rates and fewer groups generally lead to poorer performance, likely due to limited parameter capacity hindering stable learning; ❷ Increasing the number of samples or update steps has minimal impact on performance (fluctuations < 1%), but significantly increases time cost, highlighting the rationale of our flash update mechanism.

## 5.6 Efficiency & Lightweight Study (RQ5)

**Result Analysis.** We use *GWNet* as the backbone and compare ST-TTC with other test-time adaptation methods on *METR-LA* in terms of total inference time and memory usage. As shown in Figure 8, ST-TTC achieves the best overall efficiency (excluding the *GWNet* baseline), being 4.64× faster and reducing memory usage by 37.12% compared to the least efficient method, which is much smaller than the sliding size (5 min.), meeting the time requirement.

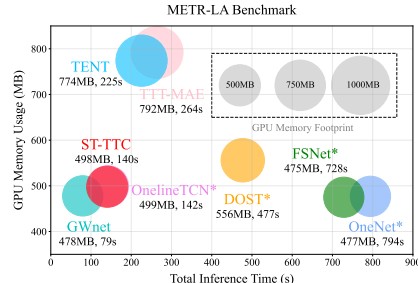

Figure 8: time and memory.

## 6 Conclusion

In this paper, we investigate the objectives of test-time computation in spatio-temporal forecasting and explore effective approaches for its implementation. We propose ST-TTC , a novel paradigm that uses a flash gradient update with streaming memory queue to learning a spectral-domain calibrator via phase-amplitude modulation, effectively addressing non-stationary errors. Extensive experiments confirm its effectiveness, universality, and flexibility. In future work, we aim to explore how to enhance the internal computational capacity of spatio-temporal foundation models during test time.

**Acknowledgments**

The first author would like to thank all the anonymous reviewers for their valuable comments and high recognition. Although he has almost lost his passion for this field and is ready to leave, he hopes that this study can still bring something new to the ST community.

This work is mainly supported by the National Natural Science Foundation of China (No. 62402414). This work is also supported by the Huawei Co., Ltd (No. TC20241023027), the Guangzhou-HKUST(GZ) Joint Funding Program (No. 2024A03J0620), Guangzhou Municipal Science and Technology Project (No. 2023A03J0011), the Guangzhou Industrial Information and Intelligent Key Laboratory Project (No. 2024A03J0628), and a grant from State Key Laboratory of Resources and Environmental Information System, and Guangdong Provincial Key Lab of Integrated Communication, Sensing and Computation for Ubiquitous Internet of Things (No. 2023B1212010007).

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

# SUPPLEMENTARY MATERIAL
## LEARNING WITH CALIBRATION: EXPLORING
## TEST-TIME COMPUTING OF SPATIO-TEMPORAL FORECASTING

## TABLE OF CONTENTS

# A   More Related Work

## A.1   Spectral Domain Learning.

Many recent forecasting models leverage spectral (Fourier or wavelet) representations to capture periodic or multiscale patterns in spatio-temporal data. For example, *PastNet* [82] integrates a Fourier-domain convolutional operator to embed physical inductive biases, achieving state-of-the-art results in weather and traffic prediction. *FourierGNN* [88] builds a learnable Fourier-graph operator that conducts graph convolutions in the frequency domain, reducing convolutional complexity from $\mathcal{O}(n^2)$ to $\mathcal{O}(n)$. Wavelet-based methods like *WDNO* [27] perform diffusion modeling in the wavelet domain to capture abrupt spatio-temporal changes and multi-resolution features. In the pure time-series setting, approaches such as *FITS* [87] interpolate in the complex Fourier domain and discard negligible high-frequency components to maintain accuracy with very few parameters, and *TimeKAN* [28] explicitly decomposes multivariate series into multiple frequency bands using Kolmogorov–Arnold networks. These works demonstrate how frequency-domain learning can improve forecasting efficiency and accuracy by isolating dominant spectral components. *Different from these methods, our method combines spectral domain feature extraction with calibration-aware test-time computation to achieve reliable and calibrated forecasts even under changing conditions.*

## A.2   Online Learning for Forecasting.

Traditional online forecasting methods include adaptive filters like Kalman filters and recursive least squares that update linear models on streaming data. Recently, deep-learning approaches have been proposed to handle nonstationarity in an online fashion. For instance, *FSNet* [57] implements a complementary "fast and slow" learning system: a fast-adapting component for sudden pattern changes and a slow memory component for repeating trends. *OneNet* [81] runs two parallel neural forecasters (one modeling temporal dependencies, one modeling cross-variable dependencies) and uses reinforcement learning to dynamically weight their predictions under concept drift. These methods continuously update model parameters or ensemble weights as new data arrive. A recent study [37] pointed out the information leakage problem of previous online time series prediction methods, where the model makes predictions and then evaluates them based on the historical time steps that have been back-propagated for parameter updates. By redefining the setting to focus on predicting unknown future steps and evaluating unobserved data points, they propose a two-stream framework for online prediction, DSOF, which is conceptually similar to previous methods, generating predictions in a coarse-to-fine manner through a teacher-student model. *Compared with these methods, we focus on the more difficult spatio-temporal predictions while not requiring complex network architecture design. Instead, we propose a calibration-aware framework that focuses on adjusting predictions online instead of learning predictions.*

## A.3   Test-Time Adaptation.

Recent test-time adaptation techniques can be grouped by their adaptation strategy. *Entropy minimization* methods adjust a trained model to increase prediction confidence on unlabeled test data. For example, [73] propose *TENT*, which adapts model parameters by minimizing the entropy of its predictions on each test batch and updating batch-normalization layers online. *Feature alignment* methods recalibrate feature distributions using test inputs; for instance, adaptive batch-normalization techniques re-estimate BN statistics on the target data to align feature distributions without labels. *Self-supervised adaptation* uses auxiliary tasks on the test data to refine the model. Test-Time Training [20, 26, 66, 78] converts each test input into a self-supervised learning problem (*e.g.* predicting image rotations) and updates model parameters before making a prediction. Similarly, SHOT [45] freezes the source classifier and updates the feature extractor on unlabeled target data using pseudo-labeling and information maximization. Each of these paradigms improves generalization under distribution shift without access to target labels. *In contrast, unlike these methods that exploit self-supervisory information, our spatio-temporal prediction setting can use labels from historical test information, enabling explicit optimization of the objective at test time, ensuring real-time adaptivity.*

# B Theoretical Analysis

## B.1 Approximate Bound on Output Perturbation

**Theorem 1** (Approximate Bound on Output Perturbation). *Let $Y \in \mathbb{C}^{B \times N \times M}$ be the original frequency-domain representation of the backbone's prediction $y \in \mathbb{R}^{B \times N \times T}$, and $y' \in \mathbb{R}^{B \times N \times T}$ be the calibrated output. Suppose the amplitude and phase modulation parameters satisfy $|\lambda_g^\alpha| \leq \epsilon_\alpha$ and $|\lambda_g^\phi| \leq \epsilon_\phi$ for all groups $g \in \{1, \ldots, G\}$. Then, the $\ell_2$-norm of the calibration error satisfies:*

$$\|y' - y\|_2 \leq (\epsilon_\alpha + \epsilon_\phi)\|Y\|_2,$$

*where $\|Y\|_2$ is the $\ell_2$-norm of $Y$.*

*Proof.* Let $\Delta Y = Y' - Y$ denote the frequency-domain perturbation. For each group $g$, the calibrated spectrum is $Y_g' = A_g(1 + \lambda_g^\alpha)e^{j(P_g + \lambda_g^\phi)}$. Expanding $Y_g'$ around $\lambda_g^\alpha = 0, \lambda_g^\phi = 0$, we approximate:

$$Y_g' \approx Y_g \left(1 + \lambda_g^\alpha + j\lambda_g^\phi\right),$$

where higher-order terms (*e.g.*, $\lambda_g^\alpha \lambda_g^\phi$) are neglected under small $\epsilon_\alpha, \epsilon_\phi$. Thus, the perturbation is:

$$\Delta Y_g \approx Y_g(\lambda_g^\alpha + j\lambda_g^\phi).$$

The $\ell_2$-norm of $\Delta Y$ is bounded by:

$$\|\Delta Y\|_2^2 = \sum_{g=1}^{G} \sum_{f \in \text{Group } g} |\Delta Y_{g,f}|^2 \leq \sum_{g=1}^{G}(\epsilon_\alpha^2 + \epsilon_\phi^2) \sum_{f \in \text{Group } g} |Y_{g,f}|^2 = (\epsilon_\alpha^2 + \epsilon_\phi^2)\|Y\|_2^2.$$

By Parseval's theorem [55], $\|y' - y\|_2 = \|\Delta Y\|_2$, hence:

$$\|y' - y\|_2 \leq \sqrt{\epsilon_\alpha^2 + \epsilon_\phi^2}\|Y\|_2 \leq (\epsilon_\alpha + \epsilon_\phi)\|Y\|_2.$$

$\square$

**Remark 1.** *This theorem guarantees that the calibration-induced perturbation is sub-linearly bounded by the modulation parameters $\epsilon_\alpha, \epsilon_\phi$. By constraining these parameters (e.g., via regularization during test-time adaptation), SD-Calibrator ensures the calibrated output does not deviate excessively from the original prediction, thereby avoiding overfitting to transient noise. The group-wise parameterization further reduces the effective degrees of freedom (from $\mathcal{O}(NM)$ to $\mathcal{O}(NG)$), inherently limiting the risk of over-parameterization.*

## B.2 Controlled Descent on Streaming Memory Queues

**Assumption 1** (Lipschitz Continuous Gradient of the Loss). *The loss function $L_k(\lambda) = \mathcal{L}(g_\lambda(f_\theta(X_o^{(k)})), Y_o^{(k)})$ is differentiable with respect to $\lambda$, and its gradient $\nabla_\lambda L_k(\lambda)$ is Lipschitz continuous with constant $L_c > 0$. That is, for any $\lambda_a, \lambda_b$:*

$$\|\nabla_\lambda L_k(\lambda_a) - \nabla_\lambda L_k(\lambda_b)\|_2 \leq L_c\|\lambda_a - \lambda_b\|_2$$

*According to the descent Lemma [50], this implies:*

$$L_k(\lambda_b) \leq L_k(\lambda_a) + \langle \nabla_\lambda L_k(\lambda_a), \lambda_b - \lambda_a \rangle + \frac{L_c}{2}\|\lambda_b - \lambda_a\|_2^2$$

**Assumption 2** (Bounded Gradient). *The norm of the gradient of the loss function with respect to the calibrator parameters $\lambda$ is bounded for any sample $(X_o^{(k)}, Y_o^{(k)})$ from the queue and any reasonable parameter set $\lambda_k$:*

$$\|\nabla_\lambda L_k(\lambda_k)\|_2 \leq G_{max}$$

*for some constant $G_{max} > 0$.*

This is a common assumption, especially if the output of the calibrator and the true labels are within a certain range, and the calibrator $g_\lambda$ is well-behaved.

**Proposition 2** (Controlled Descent on Streaming Memory Queues). *Let the above assumptions hold. For the $k$-th update step using the dequeued sample pair $(X_o^{(k)}, Y_o^{(k)})$, if the learning rate $\eta$ satisfies $0 < \eta < \frac{2}{L_c}$, then the single gradient descent step on the SD-Calibrator parameters $\lambda$ ensures a decrease in the loss function for that specific sample:*

$$L_k(\lambda_{k+1}) \leq L_k(\lambda_k) - \eta \left( 1 - \frac{L_c \eta}{2} \right) \|\nabla_\lambda L_k(\lambda_k)\|_2^2$$

*Furthermore, the change in the calibrator parameters is bounded:*

$$\|\lambda_{k+1} - \lambda_k\|_2 \leq \eta G_{max}$$

*Proof.* Let $L_k(\lambda) = \mathcal{L}(g_\lambda(f_\theta(X_o^{(k)})), Y_o^{(k)})$ be the loss for the $k$-th dequeued sample. The parameter update rule is $\lambda_{k+1} = \lambda_k - \eta \nabla_\lambda L_k(\lambda_k)$.

From Assumption 1, we have:

$$L_k(\lambda_{k+1}) \leq L_k(\lambda_k) + \langle \nabla_\lambda L_k(\lambda_k), \lambda_{k+1} - \lambda_k \rangle + \frac{L_c}{2} \|\lambda_{k+1} - \lambda_k\|_2^2$$

Substitute $\lambda_{k+1} - \lambda_k = -\eta \nabla_\lambda L_k(\lambda_k)$:

$$L_k(\lambda_{k+1}) \leq L_k(\lambda_k) + \langle \nabla_\lambda L_k(\lambda_k), -\eta \nabla_\lambda L_k(\lambda_k) \rangle + \frac{L_c}{2} \| - \eta \nabla_\lambda L_k(\lambda_k) \|_2^2$$

$$L_k(\lambda_{k+1}) \leq L_k(\lambda_k) - \eta \|\nabla_\lambda L_k(\lambda_k)\|_2^2 + \frac{L_c \eta^2}{2} \|\nabla_\lambda L_k(\lambda_k)\|_2^2$$

Factor out $\|\nabla_\lambda L_k(\lambda_k)\|_2^2$:

$$L_k(\lambda_{k+1}) \leq L_k(\lambda_k) - \eta \left( 1 - \frac{L_c \eta}{2} \right) \|\nabla_\lambda L_k(\lambda_k)\|_2^2$$

For the loss to decrease (or stay the same if gradient is zero), we require the term $\eta \left( 1 - \frac{L_c \eta}{2} \right) \|\nabla_\lambda L_k(\lambda_k)\|_2^2 \geq 0$. Since $\eta > 0$ and $\|\nabla_\lambda L_k(\lambda_k)\|_2^2 \geq 0$, we need $\left( 1 - \frac{L_c \eta}{2} \right) > 0$. This implies $1 > \frac{L_c \eta}{2}$, so $\frac{2}{L_c} > \eta$. Thus, if $0 < \eta < \frac{2}{L_c}$, the loss $L_k(\lambda_{k+1})$ on the sample $(X_o^{(k)}, Y_o^{(k)})$ is strictly reduced if $\nabla_\lambda L_k(\lambda_k) \neq 0$.

For the bound on parameter change:

$$\|\lambda_{k+1} - \lambda_k\|_2 = \| - \eta \nabla_\lambda L_k(\lambda_k) \|_2 = \eta \|\nabla_\lambda L_k(\lambda_k)\|_2$$

Using Assumption 2, $\|\nabla_\lambda L_k(\lambda_k)\|_2 \leq G_{max}$:

$$\|\lambda_{k+1} - \lambda_k\|_2 \leq \eta G_{max}$$

This completes the proof. $\qquad\square$

**Remark 2.** *The proposition demonstrates that each single-step update is not arbitrary but moves the SD-Calibrator's parameters $\lambda$ in a direction that reduces the prediction error on the specific historical sample $(X_o^{(k)}, Y_o^{(k)})$ used for the update, provided the learning rate $\eta$ is chosen appropriately (i.e., small enough, specifically $\eta < 2/L_c$). The condition on $\eta$ ensures that the update step does not overshoot. The second part, $\|\lambda_{k+1} - \lambda_k\|_2 \leq \eta G_{max}$, shows that the magnitude of change in the parameters $\lambda$ during each update is bounded. This is crucial for preventing the calibrator from experiencing excessively large or erratic parameter shifts from one step to the next, which could lead to instability or overfitting to noisy individual samples.*

## C  Method Details

### C.1  Spectral Domain Calibrator

**Algorithm Workflow.** We summarize the algorithm workflow of Section 4.1 in Algorithm 1.

**Algorithm Pseudo-code.** We further present Algorithm 1 in the form of pytorch pseudo code in Algorithm 2 for easy understanding.

---

**Algorithm 1** Spectral Domain Calibrator

---

**Require:** Pre-trained backbone $f_\theta$, Test input $x$, Horizon length $T$, Number of nodes $N$, Groups $G$
**Ensure:** Calibrated output $\hat{y}^{\text{cal}}$
  1: Get the backbone predictions: $\hat{y} = f_\theta(x) \in \mathbb{R}^{N \times T}$
  2: Compute $M \leftarrow \frac{T}{2} + 1$
    ▷ **I: Spatial-aware Decomposition**
  3: Apply real-to-complex FFT along time dimension for each node: $Y_f \leftarrow \text{rFFT}(\hat{y}) \in \mathbb{C}^{N \times M}$
  4: Decompose: $A \leftarrow |Y_f|, \quad P \leftarrow \angle Y_f$
    ▷ **II: Group-wise Modulation**
  5: **for** $g = 1, \ldots, G$ **do**
  6:   Get group index: $start \leftarrow (g-1)\lfloor M/G \rfloor + 1, \quad end \leftarrow \begin{cases} M & g = G \\ g \lfloor M/G \rfloor & \text{otherwise} \end{cases}$
  7:   Get learnable offsets: $\lambda_g^\alpha \in \mathbb{R}^{N \times 1}, \; \lambda_g^\phi \in \mathbb{R}^{N \times 1}$
  8:   Modulate group-slice: $A_g' \leftarrow A[:, start : end] \odot (1 + \lambda_g^\alpha), \quad P_g' \leftarrow P[:, start : end] + \lambda_g^\phi$
  9:   Reconstruct slice: $Y_f'[:, start : end] \leftarrow A_g' \odot e^{j\, P_g'}$
 10: **end for**
    ▷ **III: Inverse Transform**
 11: Inverse FFT: $\hat{y}^{\text{cal}} \leftarrow \text{irFFT}(Y_f') \in \mathbb{R}^{N \times T}$
 12: **return** $\hat{y}^{\text{cal}}$

---

### C.2    Lightning Gradient Update

**Algorithm Pseudo-code.** We summarize the algorithm workflow of Section 4.2 in Algorithm 3.

**Algorithm Workflow.** We further present Algorithm 3 in the form of pytorch pseudo code in Algorithm 4 for easy understanding.

## D    Experimental Details

### D.1    Datasets Details

Our experiments are carried out on 14 real-world datasets from diffrent domain. The statistics of these spatio-temporal datasets are shown in Table 5.

We follow the conventional practice [40] to define the graph topology for all spatio-temporal datasets except Know-Air. Specifically, we construct the adjacency matrix $A$ for each dataset using a threshold Gaussian kernel, defined as follows:

$$A_{[ij]} = \begin{cases} \exp\left(-\frac{d_{ij}^2}{\sigma^2}\right) & \text{if } \exp\left(-\frac{d_{ij}^2}{\sigma^2}\right) \geq r \text{ and } i \neq j \\ 0 & \text{otherwise} \end{cases}$$

where $d_{ij}$ represents the distance between sensors $i$ and $j$, $\sigma$ is the standard deviation of all distances, and $r$ is the threshold. We follow the recommended parameter settings in all corresponding papers.

For the KnowAir dataset, we follow the original paper [79] and calculate the correlation between nodes to construct the adjacency matrix. Intuitively, most aerosol pollutants are distributed within a certain range above the ground. In addition, the mountains along the two cities will hinder the transmission of pollutants to the $PM_{2.5}$ direction. Based on these intuitions, we constrain the weights in the adjacency matrix by the following formula:

$$A_{[ij]} = H(d_\theta - d_{ij}) \cdot H(m_\theta - m_{ij}), \quad \text{where}$$

$$d_{ij} = ||\rho_i - \rho_j||, \quad m_{ij} = \sup_{\lambda \in (0,1)} \{h(\lambda \rho_i + (1-\lambda)\rho_j) - \max\{h(\rho_i), h(\rho_j)\}\},$$

where $\rho_i$ is the location (latitude, longitude) of node $i$, $h(\rho)$ is the height of location $\rho$, and $|| \cdot ||$ is the L2-norm of the vector. $H(\cdot)$ is the Heaviside step function, where $H(x) = 1$ if and only if $x > 0$. $d_\theta$ and $m_\theta$ are the distance and altitude thresholds, respectively. Specifically, we also set the distance threshold $d_\theta = 300$ km and the altitude threshold $m_\theta = 1200$ meters.

**Algorithm 2** PyTorch-style pseudocode: SD-Calibrator Class

```python
class SD_Calibrator(nn.Module):
    """
    Spectral Domain Calibrator with Phase-Amplitude Modulation
    """

    def __init__(self, num_nodes, freq_bins, groups=4):
        """
        Args:
            num_nodes: number of spatial nodes (N)
            freq_bins: number of frequency bins (M = T // 2 + 1)
            groups: number of frequency groups (G)
        """
        super().__init__()
        self.groups = groups
        self.group_size = freq_bins // groups

        # Learnable offsets for amplitude and phase: (G, N, 1)
        self.lambda_amp = nn.Parameter(
            torch.zeros(groups, num_nodes, 1)
        )
        self.lambda_phi = nn.Parameter(
            torch.zeros(groups, num_nodes, 1)
        )

    def forward(self, y_pred):
        """
        Args:
            y_pred: prediction from backbone, shape (B, 1, N, T)
            B defaults to 1, because only one sample can be tested
        Returns:
            calibrated prediction, shape (B, 1, N, T)
        """
        B, _, N, T = y_pred.shape
        y = y_pred[:, 0]                        # (B, N, T)

        Yf = torch.fft.rfft(y, dim=-1)    # (B, N, M)
        A = torch.abs(Yf)
        P = torch.angle(Yf)

        Yf_corr = torch.zeros_like(Yf)
        for g in range(self.groups):
            start = g * self.group_size
            if g == self.groups - 1
              end = T // 2 + 1
            else
              end = (g + 1) * self.group_size

            lam_a = self.lambda_amp[g].unsqueeze(0)  # (1, N, 1)
            lam_p = self.lambda_phi[g].unsqueeze(0)

            A_g = A[:, :, start:end] * (1 + lam_a)
            P_g = P[:, :, start:end] + lam_p

            Yf_corr[:, :, start:end] = A_g * torch.exp(1j * P_g)

        y_time = torch.fft.irfft(Yf_corr, n=T, dim=-1)
        return y_time.unsqueeze(1)                    # (B, 1, N, T)
```

**Algorithm 3** Flash Gradient Update Mechanism

---

**Require:** Test spatio-temporal sample stream $\{x_t\}_{t=1}^B$, Pre-trained backbone $f_\theta$, Streaming memory queue $\mathcal{Q}$, Queue size $T$ (equal to horizon length)

**Ensure:** Spectral domain calibrator $g_\theta$, Prediction collection of test samples $\{\hat{y}_t^{cal}\}_{t=1}^B$

1: Initialize Calibrator module $g_\theta = (\lambda^\alpha, \lambda^\phi)$, empty queue $\mathcal{Q}$
2: **for** each timestep $t = 1, 2, ...$ **do**
3:     Receive $x_t$, compute default prediction $\hat{y}_t = f_\theta(x_t)$
    ▷ *I: Streaming Memory Queue*
4:     Use Algorithm 1 to obtain the calibration results: $\hat{y}_t^{cal} = g_\theta(\hat{y}_t^{cal}; \lambda)$
5:     Record ground truth: $y_t$ (collected by the value of $x_t$, available $T$ time steps in the future)
6:     $\mathcal{Q}$.enqueue($(x_t, y_t)$)
    ▷ *II: Flash Gradient Update*
7:     **if** len($\mathcal{Q}$) > $T$ **then**
8:         $(x_o, y_o) = \mathcal{Q}$.dequeue()
9:         Use Algorithm 1 to obtain the calibration results: $\hat{y}_o^{cal} = f_\theta(x_o)$
10:        Update: $\lambda \leftarrow \lambda - \eta\nabla_\lambda L(y_o, \hat{y}_o^{cal})$
11:     **end if**
12: **end for**
13: **return** $g_\theta, \{\hat{y}_t^{cal}\}_{t=1}^B$

---

**Algorithm 4** PyTorch-style pseudocode: Flash Gradient Update Function

---

```python
def st_ttc_test(self, test_loader, node_num, T, groups):
    """
    Flash Gradient Update with Streaming Memory Queue
    """
    SDC = SD_Calibrator(node_num, T//2+1, groups).to(self.device)
    optimizer = torch.optim.Adam(SDC.parameters(), lr=1e-4)
    loss_fn = self._select_criterion()
    SMQ, preds = Queue(maxsize=T), []

    for x, y in test_loader:
        x, y = x.to(self.device), y.to(self.device)
        with torch.no_grad():
            y_pred = self.model(x)
            y_corr = SDC(y_pred)

        # Use y_corr for inference
        y_corr = self.scaler.inverse_transform(y_corr)
        preds.append(y_corr.cpu().detach().numpy())

        SMQ.put((x, y))
        if SMQ.full():
            x_old, y_old = SMQ.get()
            with torch.no_grad():
                y_pred_old = self.model(x_old)

            SDC.train()
            y_corr_old = SDC(y_pred_old)
            y_corr_old = self.scaler.inverse_transform(y_corr_old)

            loss = loss_fn(y_corr_old, y_old)
            loss.backward()
            optimizer.step()
            optimizer.zero_grad()
            SDC.eval()
    return preds
```

Table 5: Summary of datasets used for our experiments. Degree: the average degree of each node. Data Points: multiplication of nodes and frames. M: million ($10^6$).

| Source | Dataset | Nodes | Time Range | Frames | Sampling Rate | Data Points |
|---|---|---|---|---|---|---|
| | PEMS03 | 358 | 09/01/2018 – 11/30/2018 | 26,208 | 5 minutes | 9.38M |
| [65] | PEMS04 | 307 | 01/01/2018 – 02/28/2018 | 16,992 | 5 minutes | 5.22M |
| | PEMS07 | 883 | 05/01/2017 – 08/06/2017 | 28,224 | 5 minutes | 24.92M |
| | PEMS08 | 170 | 07/01/2016 – 08/31/2016 | 17,856 | 5 minutes | 3.04M |
| [39] | UrbanEV | 275 | 09/01/2022 – 02/28/2023 | 4344 | 1 hour | 1.19M |
| [79] | Know-Air | 184 | 01/01/2015 – 12/31/2018 | 11688 | 3 hours | 2.15M |
| [40] | METR-LA | 207 | 03/01/2012 – 06/27/2012 | 34,272 | 5 minutes | 7.09M |
| | CA | 8,600 | 01/01/2019 – 12/31/2019 | 35,040 | 15 minutes | 30.13M |
| LargeST [48] | GLA | 3,834 | 01/01/2019 – 12/31/2019 | 35,040 | 15 minutes | 13.43M |
| | GBA | 2,352 | 01/01/2019 – 12/31/2019 | 35,040 | 15 minutes | 8.87M |
| | SD | 716 | 01/01/2019 – 12/31/2020 | 70,080 | 15 minutes | 5.02M |
| | Air-Stream | 1087 → 1154 → 1193 → 1202 | 01/01/2016 - 12/31/2019 | 34065 | 1 hour | 15.79M |
| [10] | PEMS-Stream | 655 → 715 → 786 → 822 → 834 → 850 → 871 | 07/10/2011 - 09/08/2017 | 61,992 | 5 minutes | 34.30M |
| | Energy-Stream | 103 → 113 → 122 → 134 | Unknown (245 days) | 34,560 | 10 minutes | 1.63M |

## D.2  Baseline Details

In our paper, we cover various spatio-temporal forecasting methods under various learning paradigms. The following is a classification and brief introduction of these advanced methods:

**Classical Learning Methods for Spatio-Temporal Forecasting.**

- *STAEformer* [47]: *STAEformer* is a spatial-temporal adaptive embedding transformer that makes vanilla transformer state-of-the-art for spatio-temporal forecasting. It introduces a novel architecture to effectively capture the dynamic spatial and temporal dependencies in spatio-temporal data. https://github.com/XDZhelheim/STAEformer

- *STTN* [86]: *STTN* is a spatial-temporal transformer network designed for traffic flow forecasting. It leverages dynamic directed spatial dependencies and long-range temporal dependencies to enhance the accuracy of long-term traffic predictions. https://github.com/xumingxingsjtu/STTN

- *GWNet* [83]: *GWNet* is a graph wavenet model for deep spatial-temporal graph modeling. It effectively captures the complex spatial and temporal patterns in spatio-temporal data using a combination of graph convolutional networks and dilated causal convolutions. https://github.com/nnzhan/Graph-WaveNet

- *STGCN* [90]: *STGCN* is a spatio-temporal graph convolutional network framework for traffic forecasting. It integrates graph convolutional networks with temporal convolutional networks to model the spatial and temporal dependencies in traffic data. https://github.com/hazdzz/stgcn

- *STID* [62]: *STID* is a simple yet effective baseline for spatio-temporal forecasting. It addresses the indistinguishability of samples in spatial and temporal dimensions by attaching spatial and temporal identity information, achieving competitive performance with concise and efficient models. https://github.com/GestaltCogTeam/STID

- *ST-Norm* [15]: *ST-Norm* is a method that applies spatial and temporal normalization for multivariate time series forecasting. It enhances the performance of forecasting models by normalizing the spatial and temporal features of the data. https://github.com/JLDeng/ST-Norm

**Efficient Learning Methods for Large-Scale Spatio-Temporal Forecasting.**

- *PatchSTG* [17]: *PatchSTG* is an attention-based dynamic spatial modeling method that uses irregular spatial patching for efficient large-scale spatio-temporal forecasting. It reduces computational complexity by segmenting large-scale inputs into balanced and non-overlapped patches, capturing local and global spatial dependencies effectively. https://github.com/lmissher/patchstg

**OOD Learning Methods for Spatio-Temporal Forecasting.**

- *STONE* [68]: *STONE* is a state-of-the-art spatio-temporal OOD learning framework that effectively models spatial heterogeneity and generates temporal and spatial semantic graphs. It introduces a graph perturbation mechanism to enhance the model's environmental modeling capability for better generalization. https://github.com/PoorOtterBob/STONE-KDD-2024

**Continual Learning Methods for Spatio-Temporal Forecasting.**

- *EAC* [10]: *EAC* is a state-of-the-art method for exploring the rapid adaptation of models in the face of dynamic spatio-temporal graph changes during supervised finetuning. It follows the principles of expand and compress to address the challenges of retraining models over new data and catastrophic forgetting. https://github.com/Onedean/EAC

- *STKEC* [70]: *STKEC* is a continual learning framework for traffic flow prediction on expanding traffic networks. It introduces a pattern bank to store representative network patterns and employs a pattern expansion mechanism to incorporate new patterns from evolving networks without requiring historical graph data. https://github.com/wangbinwu13116175205/STKEC

In addition to these advanced spatio-temporal forecasting models, we also cover various competitive baselines that learn with test information, mainly in the following three categories:

**Popular test-time training methods**

- *TTT-MAE* [20]: *TTT-MAE* is a test-time training method that uses masked autoencoders to adjust the model during inference. It helps improve the performance of the model on unseen data by effectively utilizing test-time information. We adapt it to the backbone model of the spatiotemporal network, which is divided into a feature extractor and a prediction head as well as a self-supervisory head. https://github.com/Rima-ag/TTT-MAE

- *TENT* [73]: *TENT* is a method for adjusting the model at test time by normalizing the activation function to reduce the offset between the training distribution and the test distribution. It enhances the generalization ability of the model without retraining on labeled test data. Although it is theoretically designed mainly for the cross entropy loss function, that is, classification tasks, we can still directly apply it to our prediction scenarios. https://github.com/DequanWang/tent

**Classical online time series forecasting methods**

- *OnlineTCN* [105]: *OnlineTCN* is an online learning method based on a time convolutional network. It can adapt to new data sequentially and is very suitable for real-time prediction applications where data arrives continuously. https://github.com/locuslab/TCN

- *FSNet* [57]: *FSNet* proposes a fast and slow learning network for online time series prediction that can handle both sudden changes and repeated patterns. In particular, *FSNet* improves on a slowly learning backbone by dynamically balancing fast adaptation to recent changes and retrieval of similar old knowledge. *FSNet* implements this mechanism through the interaction between two complementary components of the adapter to monitor each layer's contribution to missing events, and an associative memory that supports remembering, updating, and recalling repeated events. https://github.com/salesforce/fsnet

- *OneNet* [81]: *OneNet* dynamically updates and combines two models, one focusing on modeling dependencies across time dimensions and the other focusing on cross-variable dependencies. The approach integrates reinforcement learning-based methods into a traditional online convex programming framework, allowing the two models to be linearly combined with dynamically adjusted weights, thereby addressing the main drawback of classical online prediction methods that are slow to adapt to concept drift. https://github.com/yfzhang114/OneNet

**Advanced spatio-temporal forecasting methods using test information.**

- *CompFormer* [100]: *CompFormer* proposes a test-time compensated representation learning framework, including a spatiotemporal decomposed database and a multi-head spatial transformer model. The former component explicitly separates all training data along the time dimension according to periodic features, while the latter component establishes connections between recent observations and historical sequences in the database through a spatial attention matrix. This enables it to transfer robust features to overcome abnormal events

- *DOST* [72]: *DOST* proposes a novel online continuous learning framework tailored to the characteristics of spatiotemporal data. *DOST* adopts an adaptive spatiotemporal network equipped with variable independent adapters to dynamically address the unique distribution changes of each urban location. In addition, to adapt to the gradual nature of these transformations, a wake-sleep learning strategy is used, which intermittently fine-tunes the adapters during the online stage to reduce computational overhead.

### D.3 Protocol Details

**Metrics Detail.** We use different metrics such as MAE, RMSE, and MAPE. Formally, these metrics are formulated as following:

$$\text{MAE} = \frac{1}{n}\sum_{i=1}^{n}|y_i - \hat{y}_i|, \quad \text{RMSE} = \sqrt{\frac{1}{n}\sum_{i=1}^{n}(y_i - \hat{y}_i)^2}, \quad \text{MAPE} = \frac{100\%}{n}\sum_{i=1}^{n}\left|\frac{\hat{y}_i - y_i}{y_i}\right|$$

where $n$ represents the indices of all observed samples, $y_i$ denotes the $i$-th actual sample, and $\hat{y}_i$ is the corresponding prediction.

**Parameter Detail.** For the hyper-parameter settings of all baseline methods, we follow the parameter settings recommended by the corresponding references. For our paper, except for the robustness study section, all other experimental hyper-parameters are set uniformly: the learning rate $lr$ is 1e-4, and the number of groups $m$ is set to 4. All experiments are conducted on a Linux server equipped with a 1 × AMD EPYC 7763 128-Core Processor CPU (256GB memory) and 4 × NVIDIA RTX A6000 (48GB memory) GPUs. To carry out benchmark testing experiments, all baselines are set to run for a duration of 100∼150 epochs by default (depends on the corresponding paper), with specific timings contingent upon the method with early stop mechanism. The number of early stopping steps is set to 10.

## E More Results

### E.1 Complete Results Table

We provide complete information of the experimental tables in the main text as Table 6, 7, 8

### E.2 Visualization Case

We provide more visualization examples of test set predictions to illustrate the effectiveness of our calibration, as shown in Figure 9

## F More Discussion

### F.1 Distinction between Spatio-Temporal Forecasting and Long-Term Time Series

As stated in the paper, our work adheres to the common settings of previous short-term and long-term spatio-temporal forecasting studies (*e.g.*, $12 \rightarrow 12, 24 \rightarrow 24$) [59, 61]. However, we are fully aware of the $96 \rightarrow 96 - 720$ settings prevalent in current long-term time series forecasting. We wish to share our insights regarding this:

❶ There has been a long-standing debate concerning the long-term predictability of time series [4, 8], which we do not intend to overly critique here. Nevertheless, it is important to note that in spatio-temporal forecasting, specifically in traffic flow theory research, previous studies [101] utilizing

Table 6: Performance comparison of different models w/ and w/o `ST-TTC` in the few-shot scenario.

| Models | | Transformer-based | | | | Graph-based | | | | MLP-based | | | |
|---|---|---|---|---|---|---|---|---|---|---|---|---|---|
| | | STAEformer [47] | | STTN [86] | | GWNet [83] | | STGCN [90] | | STID [62] | | ST-Norm [15] | |
| w/ `ST-TTC` | | ✗ | ✓ | ✗ | ✓ | ✗ | ✓ | ✗ | ✓ | ✗ | ✓ | ✗ | ✓ |
| PEMS-03 | MAE | 23.57±0.90 | **22.96**±0.93 | 21.32±0.93 | **21.15**±0.92 | 21.70±0.98 | **21.43**±0.92 | 21.79±0.50 | **21.28**±0.54 | 21.81±0.24 | **21.57**±0.24 | 21.13±0.31 | **20.74**±0.28 |
| | RMSE | 37.90±1.41 | **37.10**±1.59 | 34.19±1.59 | **33.87**±1.53 | 34.52±1.44 | **34.28**±1.40 | 34.82±0.92 | **34.08**±0.89 | 35.58±0.53 | **35.18**±0.52 | 33.76±0.45 | **33.07**±0.35 |
| | MAPE(%) | 21.49±1.21 | **21.23**±1.03 | 21.14±1.35 | **21.13**±1.28 | 21.30±1.14 | **19.70**±0.57 | 22.85±0.76 | **21.59**±0.77 | 21.46±0.86 | **21.27**±0.73 | 22.57±3.13 | **22.04**±2.22 |
| PEMS-04 | MAE | 35.10±4.25 | **34.57**±4.08 | 29.76±0.37 | **29.44**±0.31 | 33.22±1.86 | **32.87**±1.95 | 29.97±0.81 | **29.66**±0.83 | 29.64±0.67 | **29.49**±0.65 | 30.66±0.11 | **30.31**±0.14 |
| | RMSE | 50.94±4.86 | **50.23**±4.59 | 44.17±0.75 | **43.89**±0.81 | 50.09±2.56 | **49.73**±2.73 | 45.96±1.24 | **45.47**±1.28 | 44.81±1.13 | **44.62**±1.10 | 45.86±0.50 | **45.33**±0.49 |
| | MAPE(%) | 23.55±3.28 | **23.40**±3.26 | 23.51±1.27 | **22.54**±0.71 | 22.97±3.52 | **22.43**±3.17 | 20.80±1.19 | **20.72**±1.19 | 22.90±1.77 | **22.70**±1.72 | 21.75±0.67 | **21.72**±0.60 |
| PEMS-07 | MAE | 30.45±0.47 | **29.70**±0.39 | 31.70±0.82 | **31.22**±0.69 | 33.17±0.65 | **32.82**±0.63 | 32.64±0.72 | **31.88**±0.77 | 31.42±1.00 | **30.91**±1.05 | 31.14±0.06 | **30.50**±0.05 |
| | RMSE | 47.89±0.81 | **46.89**±0.76 | 46.57±1.10 | **45.96**±0.93 | 49.83±0.59 | **49.36**±0.57 | 48.65±0.14 | **47.61**±0.05 | 47.51±0.82 | **46.71**±0.97 | 47.45±0.43 | **46.54**±0.42 |
| | MAPE(%) | 13.87±0.27 | **13.53**±0.23 | 14.58±0.02 | **14.45**±0.13 | 15.04±0.70 | **14.74**±0.58 | 17.13±1.40 | **16.28**±0.99 | 15.27±1.15 | **15.03**±1.20 | 14.60±0.67 | **14.18**±0.46 |
| PEMS-08 | MAE | 36.98±7.31 | **35.47**±6.03 | 24.17±0.42 | **23.81**±0.41 | 26.21±0.85 | **25.94**±0.97 | 25.97±0.25 | **25.31**±0.21 | 24.03±0.27 | **23.75**±0.28 | 24.34±0.09 | **24.06**±0.07 |
| | RMSE | 54.61±10.46 | **52.43**±8.37 | 36.89±0.45 | **36.61**±0.50 | 40.81±0.95 | **40.51**±1.11 | 38.53±0.15 | **37.80**±0.10 | 37.62±0.77 | **37.36**±0.77 | 37.45±0.16 | **37.20**±0.20 |
| | MAPE(%) | 27.38±9.55 | **26.19**±8.29 | 18.10±0.52 | **17.65**±0.54 | 17.00±1.10 | **16.52**±1.25 | 20.16±1.91 | **17.84**±0.88 | 15.07±0.53 | **14.43**±0.20 | 15.28±0.59 | **14.99**±0.29 |
| KnowAir | MAE | 18.48±0.50 | **18.16**±0.35 | 20.47±0.23 | **19.85**±0.23 | 19.32±0.32 | **19.04**±0.29 | 20.59±0.26 | **20.09**±0.25 | 22.58±1.20 | **21.72**±0.94 | 21.52±0.39 | **20.92**±0.36 |
| | RMSE | 27.20±0.09 | **26.97**±0.08 | 28.95±0.44 | **28.51**±0.46 | 27.79±0.13 | **27.58**±0.15 | 29.05±0.20 | **28.65**±0.21 | 30.25±1.03 | **29.69**±0.83 | 29.28±0.44 | **28.86**±0.41 |
| | MAPE(%) | 72.37±7.17 | **70.14**±4.89 | 85.39±1.82 | **81.21**±1.63 | 80.49±5.43 | **78.49**±5.40 | 84.80±2.40 | **82.13**±2.18 | 102.09±6.60 | **95.69**±5.08 | 95.24±2.26 | **91.11**±1.80 |
| UrbanEV | MAE | 3.39±0.12 | **3.36**±0.07 | 4.12±0.06 | **4.05**±0.06 | 3.92±0.09 | **3.88**±0.63 | 4.21±0.10 | **4.10**±0.10 | 3.31±0.06 | **3.24**±0.05 | 4.85±0.10 | **4.75**±0.11 |
| | RMSE | 6.15±0.21 | **6.05**±0.16 | 6.81±0.06 | **6.71**±0.05 | 6.64±0.12 | **6.59**±0.11 | 6.74±0.19 | **6.61**±0.17 | 5.51±0.10 | **5.42**±0.08 | 8.54±0.27 | **8.36**±0.27 |
| | MAPE(%) | 31.60±0.19 | **31.33**±0.47 | 38.41±1.43 | **37.32**±1.33 | 35.79±1.24 | **34.95**±1.10 | 40.93±1.48 | **39.57**±1.52 | 31.42±0.59 | **30.75**±0.50 | 43.20±1.28 | **42.26**±1.25 |

Table 7: PatchSTG with `ST-TTC` in LargeST Benchmark.

| Datasets | Methods | Horizon 3 | | | Horizon 6 | | | Horizon 12 | | | Average | | |
|---|---|---|---|---|---|---|---|---|---|---|---|---|---|
| | | MAE | RMSE | MAPE(%) | MAE | RMSE | MAPE(%) | MAE | RMSE | MAPE(%) | MAE | RMSE | MAPE(%) |
| SD | PatchSTG | 14.53 | 24.34 | 9.22 | 16.86 | 28.63 | 11.11 | 20.66 | 36.27 | 14.72 | 16.90 | 29.27 | 11.23 |
| | w/ `ST-TTC` | 14.44 | 24.07 | 8.70 | 16.66 | 28.18 | 10.93 | 20.37 | 35.68 | 14.27 | 16.72 | 28.83 | 11.11 |
| | Δ | ↓0.6% | ↓1.1% | ↓5.6% | ↓1.2% | ↓1.6% | ↓1.6% | ↓1.4% | ↓1.6% | ↓3.1% | ↓1.1% | ↓1.5% | ↓1.1% |
| GBA | PatchSTG | 16.81 | 28.71 | 12.25 | 19.68 | 33.09 | 14.51 | 23.49 | 39.23 | 18.93 | 19.50 | 33.16 | 14.64 |
| | w/ `ST-TTC` | 16.65 | 28.31 | 12.12 | 19.30 | 32.40 | 14.35 | 22.96 | 38.33 | 18.30 | 19.16 | 32.49 | 14.48 |
| | Δ | ↓1.0% | ↓1.4% | ↓1.1% | ↓1.9% | ↓2.1% | ↓1.1% | ↓2.3% | ↓2.3% | ↓3.3% | ↓1.7% | ↓2.0% | ↓1.1% |
| GLA | PatchSTG | 15.84 | 26.34 | 9.27 | 19.06 | 31.85 | 11.30 | 23.32 | 39.64 | 14.60 | 18.96 | 32.33 | 11.44 |
| | w/ `ST-TTC` | 15.78 | 26.08 | 9.15 | 18.76 | 31.29 | 11.21 | 22.86 | 38.89 | 14.35 | 18.69 | 31.78 | 11.36 |
| | Δ | ↓0.4% | ↓1.0% | ↓1.3% | ↓1.6% | ↓1.8% | ↓0.8% | ↓2.0% | ↓1.9% | ↓1.7% | ↓1.4% | ↓1.7% | ↓0.7% |
| CA | PatchSTG | 14.69 | 24.82 | 10.51 | 17.41 | 29.43 | 12.83 | 21.20 | 36.13 | 16.00 | 17.35 | 29.79 | 12.79 |
| | w/ `ST-TTC` | 14.59 | 24.61 | 10.40 | 17.14 | 28.97 | 12.51 | 20.76 | 35.38 | 15.54 | 17.10 | 29.31 | 12.53 |
| | Δ | ↓0.7% | ↓0.8% | ↓1.0% | ↓1.6% | ↓1.6% | ↓2.5% | ↓2.1% | ↓2.1% | ↓2.9% | ↓1.4% | ↓1.6% | ↓2.0% |

residual analysis have indicated that, after minimizing periodicity, the correlation between traffic data beyond one hour and the past hour's observations is significantly limited for most sensors. Furthermore, typical traffic sensor data collection frequency is approximately 5 minutes. Therefore, for practical traffic forecasting and decision-making scenarios, which usually focus on the next 1-2 hours (*i.e.*, max 24 steps), our settings are more aligned with real-world applications.

❷ Given that spatio-temporal forecasting can be considered a complex extension of time series forecasting, the additional dimension of sensor count (up to hundreds or thousands) results in an order of magnitude higher data training cost. This is a secondary reason why existing spatio-temporal forecasting often considers 12-step settings.

❸ Another notable finding is that a recent study on the accuracy law of deep time series forecasting [80], emerging subsequent to this paper, identifies a significant exponential relationship between the minimum prediction error of current deep forecasting models and the complexity of window series patterns. Adopting a Spectral-domain perspective similar to that of this paper, it defines the complexity of series patterns as the total variance of the amplitude spectrum distribution, thereby characterizing the intrinsic heterogeneity of series variations within each relevant window. This approach aligns with ours and provides a novel perspective to explain the learnability and effectiveness of the single-step gradient descent in our calibrator. While the study focuses solely on univariate time series forecasting, this limitation does not inherently undermine its relevance. It is worth noting that the study concludes that mainstream time series models have not yet reached saturation on traffic scenario-based benchmarks. However, we hold a different view, arguing that this conclusion primarily stems from the study's exclusive use of time series forecasting models, while neglecting mainstream

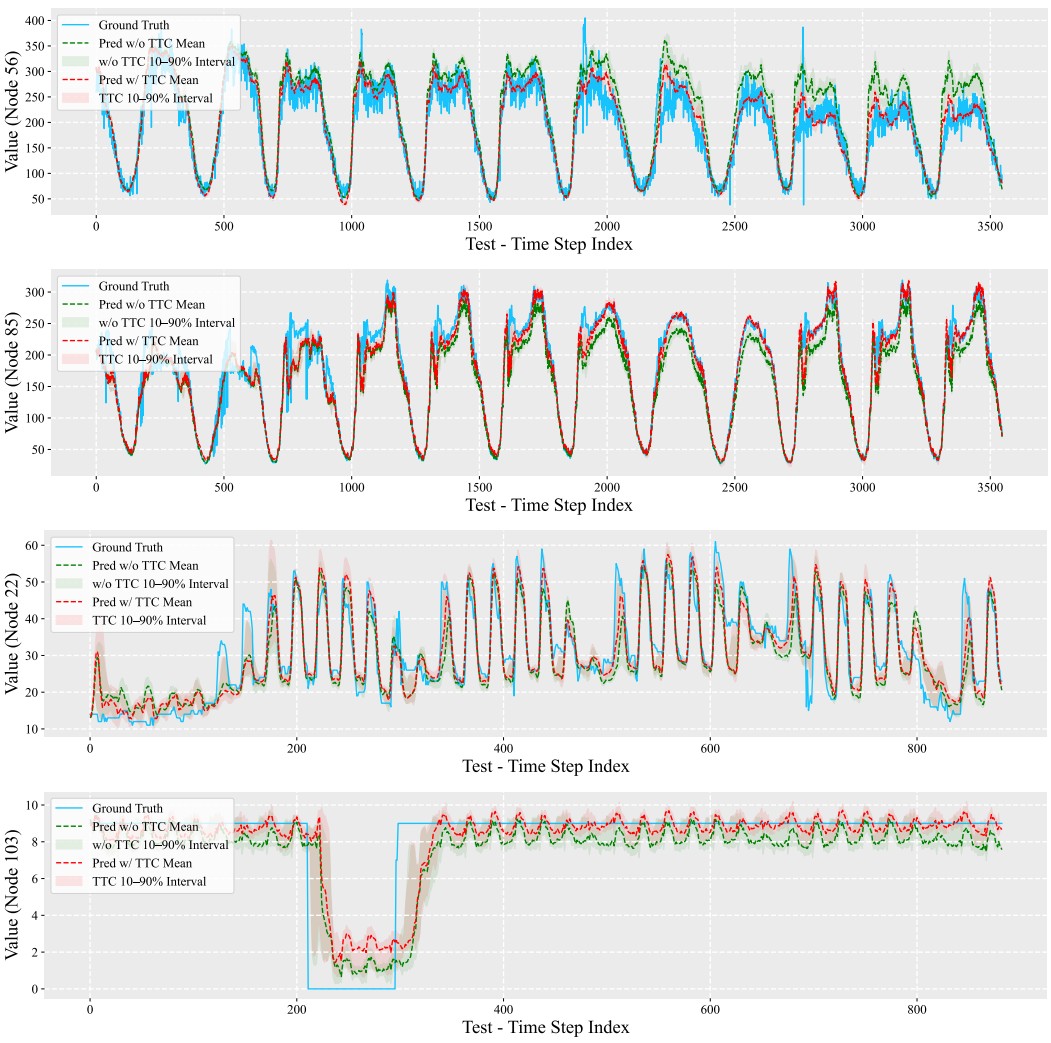

Figure 9: show case.

spatio-temporal dependency modeling models. For further in-depth discussions, we suggest that future research should address this aspect.

## F.2 Limitation

In this paper, we propose a novel paradigm for spatio-temporal forecasting: test-time computing. Although there are still many potential areas for improvement, given the superiority and generality of our `ST-TTC`, we believe this provides a pathway for future exploration of larger-scale and more effective test-time computation. While we have taken a small step in this direction, several limitations warrant attention:

❶ Our current study does not involve testing on spatio-temporal foundation models. The fundamental reason behind this is our belief that true spatio-temporal foundation models do not yet exist. Although some preliminary exploratory work has been done [92, 93, 42, 41, 91], they are far from achieving true zero-shot generalization. However, considering their future inevitability, we believe that further improving the paradigm of test-time computation, especially how to activate and scale the internal capabilities of spatio-temporal foundation models during testing, goes beyond the design philosophy of our proposed learning with calibration. Nevertheless, our experiments still provide some preliminary guidance and insights.

Table 8: Performance of spatio-temporal shift dataset SD-ratio(%) on all nodes and unknown new nodes at different spatio-temporal shift levels.

| Dataset | Horizon | Methods | All Node | | | New Node | | |
|---|---|---|---|---|---|---|---|---|
| | | | MAE | RMSE | MAPE(%) | MAE | RMSE | MAPE(%) |
| SD-10% | 12 | STONE | $40.74_{\pm4.24}$ | $58.43_{\pm3.67}$ | $45.82_{\pm9.30}$ | $46.25_{\pm7.02}$ | $66.97_{\pm7.05}$ | $47.57_{\pm8.02}$ |
| | | w/ ST-TTC | $39.41_{\pm3.82}$ | $57.44_{\pm3.67}$ | $38.02_{\pm5.19}$ | $44.65_{\pm6.69}$ | $65.06_{\pm6.95}$ | $41.85_{\pm5.87}$ |
| | | Δ | ↓3.3% | ↓1.7% | ↓17.0% | ↓3.5% | ↓2.9% | ↓12.0% |
| | Avg. | STONE | $30.18_{\pm1.19}$ | $42.83_{\pm1.38}$ | $39.44_{\pm3.03}$ | $32.79_{\pm2.77}$ | $46.68_{\pm3.87}$ | $40.12_{\pm0.30}$ |
| | | w/ ST-TTC | $28.29_{\pm1.04}$ | $41.50_{\pm1.16}$ | $28.66_{\pm1.32}$ | $30.66_{\pm2.63}$ | $44.44_{\pm3.32}$ | $30.80_{\pm1.54}$ |
| | | Δ | ↓6.3% | ↓3.1% | ↓27.3% | ↓6.5% | ↓4.8% | ↓23.2% |
| SD-15% | 12 | STONE | $35.31_{\pm0.46}$ | $52.87_{\pm0.57}$ | $34.05_{\pm1.73}$ | $43.65_{\pm0.85}$ | $66.07_{\pm1.28}$ | $30.47_{\pm1.40}$ |
| | | w/ ST-TTC | $34.86_{\pm0.15}$ | $52.25_{\pm0.54}$ | $29.80_{\pm0.79}$ | $41.93_{\pm0.62}$ | $63.17_{\pm0.29}$ | $27.70_{\pm0.03}$ |
| | | Δ | ↓1.3% | ↓1.2% | ↓12.5% | ↓3.9% | ↓4.4% | ↓9.1% |
| | Avg. | STONE | $28.45_{\pm0.05}$ | $41.30_{\pm0.26}$ | $36.02_{\pm2.91}$ | $33.20_{\pm0.69}$ | $49.19_{\pm0.97}$ | $29.77_{\pm3.00}$ |
| | | w/ ST-TTC | $26.59_{\pm0.22}$ | $39.90_{\pm0.09}$ | $24.96_{\pm1.00}$ | $30.65_{\pm0.29}$ | $46.24_{\pm0.52}$ | $22.50_{\pm0.38}$ |
| | | Δ | ↓6.5% | ↓3.4% | ↓30.7% | ↓7.7% | ↓6.0% | ↓24.4% |
| SD-20% | 12 | STONE | $36.13_{\pm0.76}$ | $53.87_{\pm0.44}$ | $34.19_{\pm1.08}$ | $41.64_{\pm1.23}$ | $63.19_{\pm2.11}$ | $37.38_{\pm3.29}$ |
| | | w/ ST-TTC | $35.08_{\pm1.08}$ | $52.37_{\pm0.67}$ | $30.55_{\pm1.13}$ | $40.10_{\pm1.41}$ | $60.77_{\pm2.07}$ | $33.70_{\pm2.73}$ |
| | | Δ | ↓2.9% | ↓2.8% | ↓10.6% | ↓3.7% | ↓3.8% | ↓9.8% |
| | Avg. | STONE | $28.86_{\pm0.13}$ | $41.72_{\pm0.14}$ | $34.89_{\pm1.15}$ | $31.46_{\pm0.95}$ | $46.06_{\pm1.60}$ | $36.35_{\pm4.23}$ |
| | | w/ ST-TTC | $26.67_{\pm0.29}$ | $39.71_{\pm0.08}$ | $24.92_{\pm0.81}$ | $28.94_{\pm0.79}$ | $43.29_{\pm1.32}$ | $26.60_{\pm2.49}$ |
| | | Δ | ↓7.6% | ↓4.8% | ↓28.6% | ↓8.0% | ↓6.0% | ↓26.8% |

❷ It is undoubtedly encouraging that our current calibration mechanism is more effective in large-scale and out-of-distribution scenarios. However, for commonly used small spatio-temporal benchmark datasets, the performance improvement is not yet significant. Therefore, how to effectively improve the performance of test-time computation on small-scale spatio-temporal datasets still requires exploration, and we reserve further improvement efforts for future research.

❸ We observed in our experiments that utilizing a larger amount of test information that is more similar to the current test sample does not significantly affect the results. This is certainly beneficial for real-time efficiency requirements. However, considering our current efficiency is already sufficiently good, further exploration is needed on how to potentially slow down the test-time computing process to make it more scalable and improve forecasting effectiveness.

## F.3 Future Work

Building upon the research direction presented in this paper, we envision future work encompassing two main aspects:

❶ Exploring how to integrate retrieval-augmented techniques to filter more effective learning samples from arbitrary external scenarios, thereby combining them with our test-time computation framework to optimize performance on small-scale datasets.

❷ Investigating the construction of real spatio-temporal foundation models that encapsulate internal compressed knowledge, and exploring how to activate this internal capability during test time.

## G  Broader Impacts

This paper aims to promote the real-world usability of spatio-temporal forecasting models. We propose a novel paradigm, namely test-time computing of spatio-temporal forecasting. This paradigm shows significant generalization, universality across multiple scenarios, multiple tasks, multiple learning paradigms, and scalability to improve performance, providing valuable insights for future research and application value for practitioners. This paper focuses mainly on scientific research and has no obvious negative impact on society.

