# OpenReview forum: "Learning with Calibration: Exploring Test-Time Computing of Spatio-Temporal Forecasting"
_NeurIPS.cc/2025/Conference — NeurIPS 2025 spotlight_

### Official Review · Reviewer_T8C2 · 2025-07-01

**Clarity:** 3
**Significance:** 3
**Originality:** 3
**Rating:** 4
**Confidence:** 4

**Summary:**

This work proposes ​ST-TTC, a novel test-time computing paradigm for spatio-temporal forecasting that dynamically corrects prediction biases during inference, and conduct effective experiments on real-world evaluation datasets.

**Questions:**

1. Video prediction is widely adopted task format for spatio-temperol prediction, especially in met and traffic domains, does ST-TTS works in video prediction tasks?

2. Apart form period shift, can ST-TTS face other OOD challenges, like corrupted input data, incident/extreme cases ?

**Ethical Concerns:**

["NO or VERY MINOR ethics concerns only"]

**Final Justification:**

This paper presents Spatio-Temporal Test-Time Computing (ST-TTC), a novel lightweight calibration framework for spatio-temporal forecasting that tackles non-stationarity and distribution shifts without modifying the backbone model or needing extra training data. The paper perform detailed experiments and is well written.

**Limitations:**

yes

**Paper Formatting Concerns:**

The paper is well formatted

**Quality:**

3

**Strengths And Weaknesses:**

Strength
1. Calibration is a light-weighted test-time training method, and a good solution for maintaining a long-term deloyed spatial-temporal prediction model.
2. Calibrate in spectral space sounds promising.
3. The method acquires universal performance gain across multiple real-world benchmarks.

Weaknesses
1. The paper discuss the difference between test-time training and other solutions for OOD prediction, but does not conduct experiments comparing these methods. The authors need to answer with experimental results: "In what case, we need to consider test-time training instead of CT or Online CT etc. ?"
2. The paper report the relative improvement ratios, but the absolute improvements seems not so significant.

---

> ### Author Response · Authors · 2025-08-01
> **Rebuttal by Authors**
>
> Dear Reviewer T8C2,
>
> We are very sorry for the slight delay in our response due to a fire at our server room yesterday.
>
> We sincerely thank you for taking the time and effort to carefully review our work. We are honored by your recognition of our hard work and have meticulously considered each of your comments, addressing them one by one.
>
> ---
>
> > ### Method-related
>
> - **`Comparison with Other OOD Solutions:`**
>     - We apologize for any misunderstanding regarding our ST-TTC method. Our approach is designed as a **general plug-in** and is orthogonal to existing OOD methods, including continual (CT) learning approaches. This means our method doesn't compete with them; rather, it can be applied on top of these models to further enhance their performance. Our goal is to maintain the inherent ability of these models to handle spatio-temporal shifts while simultaneously improving their generalization to new distribution shifts during actual deployment.
> - **`The absolute improvement does not seem significant enough:`**
>     - We agree that our method may not show significant improvements on small-scale spatio-temporal benchmarks. We clearly acknowledged this limitation in **Appendix F.1**. We must point out that the test sets for these smaller benchmarks are typically too short (spanning only days or weeks) compared to larger datasets or continual learning benchmarks (which can span months or years). This limited duration often doesn't lead to severe out-of-distribution shifts, which our method is designed to address. However, our method still provides a certain level of performance improvement, and this gain can be combined with other complex model designs to further enhance overall performance.
>     - Furthermore, in scenarios with severe spatio-temporal distribution shifts (e.g., OOD and continual spatio-temporal forecasting), the absolute improvements from our method are quite significant. For instance, we achieved an **absolute MAPE improvement of around 10%** in both OOD and continual learning settings.
>
> ---
>
> > ### Experiments-related
>
> - **`Effectiveness on Video Prediction Tasks?:`**
>     - Good question! Video prediction can be considered a special form of spatio-temporal forecasting on a grid (i.e., Euclidean vs. non-Euclidean spatial structures). However, the research community typically treats these two tasks separately. Video prediction models often use vision backbones to capture inductive biases like translation invariance, while spatio-temporal forecasting models use graph backbones to capture locality and permutation invariance. From a data perspective, there is some overlap, as traffic and meteorological data can be represented as grids similar to video inputs.
>     - To test the effectiveness of our method on this type of data, we selected two common datasets for such studies, **NYC-Taxi** and **T-Drive**, and their corresponding spatio-temporal backbone model, **PDFormer** [1]. The results are as follows:
>
> | Dataset | Metric | w/o ST-TTC | w/ ST-TTC |
> | --- | --- | --- | --- |
> | NYC-Taxi (In-Flow) | MAE | 13.18 | **12.95** |
> |  | RMSE | 21.98 | **21.74** |
> |  | MAPE (%) | 12.76 | **12.51** |
> | NYC-Taxi (Out-Flow) | MAE | 11.59 | **11.30** |
> |  | RMSE | 18.40 | **18.06** |
> |  | MAPE (%) | 12.83 | **12.46** |
> | T-Drive (In-Flow) | MAE | 17.83 | **17.01** |
> |  | RMSE | 31.61 | **30.74** |
> |  | MAPE (%) | 14.71 | **14.03** |
> | T-Drive (Out-Flow) | MAE | 17.74 | **16.97** |
> |  | RMSE | 31.50 | **30.72** |
> |  | MAPE (%) | 14.65 | **13.98** |
> - **`Handling Other OOD Challenges (input data corruption, events/corner cases) ?:`**
>     - We are happy to clarify this point. Our core objective is to focus on calibrating gradual, systematic periodic shifts while avoiding the fitting of anomalous noise from data corruption or abrupt structural changes from extreme events. _Recall that these kinds of anomalous changes are typically accidental, temporary, and non-repeatable. If our calibrator tries to adapt to past anomalies and use them to calibrate current forecasts, this will lead to negative effects._ Our proposed phase-amplitude modulation is specifically designed to prevent this exact situation.
>
> ---
> **References:**
>
> [1] PDFormer: Propagation Delay-Aware Dynamic Long-Range Transformer for Traffic Flow Prediction. AAAI, 2023.
>
> ---
>
> Thank you again for your valuable feedback. We will appropriately incorporate these detailed discussions into the final revised version. We are grateful for your guidance and suggestions!
>
>
> Best regards,
>
> Authors.

---

> > ### Comment · Reviewer_T8C2 · 2025-08-05
> >
> > Thank you for the detailed response. The clarifications and additional experiments have satisfactorily addressed my concerns. I decide to maintain my score, and increase the confidence to 4.

---

> > > ### Author Response · Authors · 2025-08-05
> > > **Thank you for your time and consideration**
> > >
> > > Dear Reviewer T8C2,
> > >
> > > We sincerely appreciate you taking the time to read our slightly delayed response. We're very pleased to hear that we've addressed all your concerns and will continue to refine the revised manuscript. Thank you for your time and guidance in reviewing our paper!
> > >
> > > Sincerely,
> > >
> > > The Authors

---

### Official Review · Reviewer_xiA6 · 2025-07-02

**Clarity:** 3
**Significance:** 3
**Originality:** 3
**Rating:** 5
**Confidence:** 4

**Summary:**

This paper introduces Spatio-Temporal Test-Time Computing (ST-TTC), a novel and lightweight calibration framework for spatio-temporal forecasting that addresses challenges such as non-stationarity and distribution shifts without modifying the backbone model or requiring additional training data. The proposed method incorporates a Spectral Domain Calibrator, which mitigates periodic biases in the frequency domain through phase-amplitude modulation, alongside a Flash Gradient Update mechanism utilizing a Streaming Memory Queue to efficiently adapt calibration parameters during inference. Extensive experiments across diverse benchmarks consistently demonstrate that ST-TTC enhances forecasting accuracy over strong baselines while incurring minimal computational overhead.

**Questions:**

The experimental evaluation is comprehensive. However, my main concern lies in the performance under OOD conditions. Since the extent of distribution shifts in the datasets is not quantified, it remains unclear how effectively the proposed method can handle truly OOD scenarios. In particular, spatio-temporal distribution shifts can manifest at both structural and temporal levels. While Figure 5 offers some insight into this, the degree of shift introduced by new nodes is not explicitly presented. It would be very helpful for the authors to clarify the following points:

1. To what extent do temporal feature distribution shifts exist in the datasets used? It would be beneficial to quantify the degree of such shifts using appropriate metrics or visualizations.

2. Does the current framework adapt to structural shifts such as changes in node homophily or the overall graph topology?

3. Does the proposed framework support dynamic graphs with evolving structures over time?

**Ethical Concerns:**

["NO or VERY MINOR ethics concerns only"]

**Final Justification:**

The authors have addressed most of my concerns regarding the effectiveness of the proposed method in the structural shift setting and have clarified my questions about the OOD setting, temporal shifts, and dynamic graphs.

**Limitations:**

yes

**Quality:**

3

**Strengths And Weaknesses:**

Strengths 1.	The paper addresses an important and practical problem in spatio-temporal modeling. Non-stationary data is prevalent and challenging, and proposing a generalizable framework to leverage test-time data is a promising direction.
2.	The use of spectral domain calibration with phase-amplitude modulation is theoretically grounded, interpretable, and empirically shown to effectively mitigate non-stationary distribution shifts to a certain extent.
3.	The experimental evaluation is thorough, covering a broad range of models, datasets, and scenarios, including few-shot, long-term, large-scale, out-of-distribution (OOD), and continual learning settings, demonstrating the method’s versatility.

⸻

Weaknesses
1.	The proposed approach focuses on periodic biases and may not generalize well to non-periodic or abrupt structural changes commonly observed in spatio-temporal data.
2.	Although the method is designed to handle OOD scenarios, the paper lacks a formal and quantitative definition of OOD in the spatio-temporal context, and the degree of distribution shifts present in the evaluated datasets remains unclear.

---

> ### Author Response · Authors · 2025-08-01
> **Rebuttal by Authors**
>
> Dear Reviewer xiA6,
>
> We are very sorry for the slight delay in our response due to a fire at our server room yesterday.
>
> Thank you for your hard work and valuable feedback. We understand your concerns and believe they stem from unnecessary misunderstandings. We are happy to clarify and address each of your questions.
>
> ---
> > ### Method-related
> - **`Generalizability to Non-Periodic or Abrupt Structural Changes:`**
>     - We are glad to clarify this point. Our core objective is to focus on calibrating gradual, systematic periodic shifts while avoiding the fitting of anomalous noise and abrupt structural changes. **Recall that these kinds of anomalous changes are typically accidental, temporary, and non-repeatable. If our calibrator tries to adapt to past anomalies and use them to calibrate current forecasts, this will lead to negative effects.** Our proposed phase-amplitude modulation is specifically designed to prevent this exact situation.
>
> ---
> > ### Experiments-related
> - **`Formalization of OOD Scenarios:`**
>     - We apologize for this misunderstanding. Our method is designed to be orthogonal to existing OOD methods, meaning we can achieve significant performance gains on top of their results. Regarding the formal definition of ST-OOD, we actually provided a general formal definition in **Table 1**, which is widely used and consistent with the ST-OOD literature [1, 2, 3].
> - **`Quantifying Distribution Shifts:`**
>     - We followed the setup of previous ST-OOD experiments [1] to simulate a significant distribution shift along the temporal dimension. Specifically, we used traffic flow data from 1 to 8, 2019 (spring and summer) as the training set and traffic flow from 11 to 12, 2020 (winter) as the test set. This setup exacerbates the temporal distribution shift across both years and seasons. Due to the limitations of the rebuttal phase, we can’t provide visual quantifications of the distribution, but metrics for this can be found in the LargeST paper [4]. We appreciate your suggestion and will consider adding more visualizations to quantify this shift in the revised version.
> - **`Adaptability to Structural Changes:`**
>     - Our method is explicitly designed to handle changes in graph structure. We addressed this issue on **line 139**, where we mentioned our use of a spatial-aware decomposition scheme to create node-level parameter calibrators. This allows our method to handle changes in graph topology by selectively loading parameters. When new nodes are added, we simply supplement the corresponding node-level calibration parameters. When nodes are removed, we don't load their parameters.
>     - Furthermore, since our method is orthogonal to existing OOD models, we can preserve their inherent ability to handle topological changes while simultaneously improving the model's generalization capabilities to unknown topologies.
>     - We evaluated performance under varying degrees of topology changes (10%, 15%, 20% node changes) on the test set in **Table 3**. We also specifically evaluated performance on new nodes in the test graph, as shown in **Table 8**, demonstrating that we significantly improve generalization on top of existing OOD models. To save you time, here is a summary of the results from Table 8:
>
> | Spatial Shift Degree | Metric | w/o ST-TTC | w/ ST-TTC |
> | --- | --- | --- | --- |
> | **10%** | MAE | 32.79 | **30.66** |
> |  | RMSE | 46.68 | **44.44** |
> |  | MAPE (%) | 40.12 | **30.80** |
> | **15%** | MAE | 33.20 | **30.65** |
> |  | RMSE | 49.19 | **46.24** |
> |  | MAPE (%) | 29.77 | **22.50** |
> | **20%** | MAE | 31.46 | **28.94** |
> |  | RMSE | 46.06 | **43.29** |
> |  | MAPE (%) | 36.35 | **26.60** |
> - **`Support for Dynamic Graphs:`**
>     - Our method is also designed to adapt to dynamic graph structures that evolve over time. This is consistent with the experimental settings of continual spatio-temporal learning [5]. In **Section 5.4**, we specifically investigated the performance improvements of our method on existing continual ST learning approaches, as detailed in **Table 4**. Our method effectively adapts to ST forecasting on dynamic graphs that evolve over time, even across periods as long as seven years.
>
> ---
>
> **References:**
>
> [1] STONE: A Spatio-temporal OOD Learning Framework Kills Both Spatial and Temporal Shifts. KDD, 2024.
>
> [2] Robust Spatio-Temporal Centralized Interaction for OOD Learning. ICML, 2025.
>
> [3] Zhang, et al. STRAP: Spatio-Temporal Pattern Retrieval for Out-of-Distribution Generalization. arXiv, 2025.
>
> [4] LargeST: A Benchmark Dataset for Large-Scale Traffic Forecasting. NeurIPS, 2023.
>
> [5] Expand and Compress: Exploring Tuning Principles for Continual Spatio-Temporal Graph Forecasting. ICLR, 2025.
>
> ---
> We believe that your concerns may stem from a potential misunderstanding of our method. We apologize for any confusion and will revise the details in the appendix more carefully. We appreciate your guidance and suggestions and welcome further discussions!
>
> Best regards,
>
> Authors.

---

> > ### Comment · Reviewer_xiA6 · 2025-08-05
> >
> > Thank you for the detailed response. Most of my concerns have been addressed, and I would appreciate seeing some quantification of OOD in the final revision.

---

> > > ### Author Response · Authors · 2025-08-06
> > > **Thank you for your time and consideration**
> > >
> > > Dear Reviewer xiA6,
> > >
> > > We sincerely appreciate your willingness to acknowledge and respond to our rebuttal. We are pleased to have addressed your concerns and will improve the final version based on your comments (add a quantification picture of OOD dataset). Thank you for your time and guidance in reviewing our paper!
> > >
> > > Sincerely,
> > >
> > > The Authors

---

### Official Review · Reviewer_N98K · 2025-07-03

**Clarity:** 4
**Significance:** 3
**Originality:** 3
**Rating:** 5
**Confidence:** 4

**Summary:**

The paper proposed a novel test-time computing paradigm for spatio-temporal forecasting, ST-TTC, where a plug-in calibration module is added between the base-model prediction and the final predictions. This calibration module improves performance at sporadic non-stationary spatio-temporal behavior of the data without expensive training. The method is evaluated on traffic, meteorological, and energy datasets. ST-TTC shows promising improvement in performance by efficiently using the computing resources.

**Questions:**

None

**Ethical Concerns:**

["NO or VERY MINOR ethics concerns only"]

**Limitations:**

1) Lack of testing on spatio-temporal foundation models
2) Does not significantly improve performance on small spatio-temporal benchmarks

**Paper Formatting Concerns:**

No paper formatting issue

**Quality:**

4

**Strengths And Weaknesses:**

Strengths:
1) The codebase is available and documented for review.
2) The introduction motivates the problem very well.
3) Figure 1 clearly depicts the difference between test-time training and online continual learning.
4) Related work is well placed.
5) The paper explains motivation, key challenges, implementation details with mathematical details, complexity, and theoretical analysis for the methodical design choices.
6) Tested on six publicly available datasets with Transformer-based, Graph-based, and MLP-based base models. Also outperforms the baselines in most scenarios.
7) The evaluation shows improvement (%) for different base-model architectures.
8) ST-TTC is efficient and robust, as shown in the results
9) Clearly mentioned the observed limitations and future directions.

Weakness:
1) Missing an Overview diagram for the methodology. I suggest including a pictorial representation of the methodology.

---

> ### Author Response · Authors · 2025-08-01
> **Rebuttal by Authors**
>
> Dear Reviewer N98K,
>
> We are very sorry for the slight delay in our response due to a fire at our server room yesterday.
>
> We sincerely thank you for taking the time and effort to carefully review our work. We are honored by your recognition of our hard work and have meticulously considered each of your comments, addressing them one by one.
>
> ---
>
> > ### Method-related
>
> - **`Performance on Small-Scale Spatio-Temporal Benchmarks:`**
>     - We agree with your assessment. As we clearly acknowledged in the limitations section (Appendix F.1), our method's effectiveness is constrained on small benchmarks. We also outlined a potential future direction: enhancing performance on these smaller datasets by incorporating our method with retrieval-augmented models for arbitrary exogenous data or variables.
>     - However, we must also point out that the test sets of small benchmarks [1] are often too short (spanning only days or weeks) compared to those of large-scale [2,3] or continual learning benchmarks [4] (which can span months or years). This limited span does not typically lead to severe out-of-distribution shifts, which our method is designed to address. Nevertheless, we still achieved a certain performance improvement. This gain, though modest, can be combined with other complex model designs to further enhance overall performance.
>     - Furthermore, in scenarios with severe spatio-temporal distribution shifts (e.g., OOD and continual spatio-temporal forecasting), the absolute improvements from our method are quite significant. For instance, we achieved an absolute MAPE improvement of around 10% in both OOD and continual learning settings.
>
> ---
>
> > ### Presentation-related
>
> - **`Lack of Method Overview Diagram:`**
>     - Good suggestion! We agree with your point. Due to the limitations of the rebuttal period, we are unable to upload images. We plan to incorporate a visual representation of our method into the final revised version.
>
> ---
>
> > ### Experiments-related
>
> - **`Lack of Testing on Spatio-Temporal Foundation Models:`**
>     - We clearly discussed our thoughts on this issue in the limitations section (Appendix F.1). However, we also agree with your feedback and have supplemented our experiments with the advanced ST-foundation model, Opencity [5]. We used the official OpenCity-mini weights and tested it on the PEMS04, PEMS07M, and PEMS08 datasets. The results are shown in the following table. Although the improvements are limited (a fact we attribute to the restricted time span of the provided benchmarks), we still achieved a noticeable gain.
>
> | Dataset | Metric | w/o ST-TTC | w/ ST-TTC |
> | --- | --- | --- | --- |
> | PEMS04 | MAE | 22.46 | 22.39 |
> |  | RMSE | 36.90 | 36.84 |
> |  | MAPE (%) | 15.68 | 15.61 |
> | PEMS07M | MAE | 4.66 | 4.61 |
> |  | RMSE | 8.55 | 8.52 |
> |  | MAPE (%) | 13.00 | 12.94 |
> | PEMS08 | MAE | 18.46 | 18.40 |
> |  | RMSE | 32.36 | 32.28 |
> |  | MAPE (%) | 15.18 | 15.16 |
>
> ---
>
> **References:**
>
> [1] Spatial-Temporal Synchronous Graph Convolutional Networks: A New Framework for Spatial-Temporal Network Data Forecasting. AAAI, 2020.
>
> [2] LargeST: A Benchmark Dataset for Large-Scale Traffic Forecasting. NeurIPS, 2023.
>
> [3] Efficient Large-Scale Traffic Forecasting with Transformers: A Spatial Data Management Perspective. KDD, 2025.
>
> [4] TrafficStream: A Streaming Traffic Flow Forecasting Framework Based on Graph Neural Networks and Continual Learning. IJCAI, 2021
>
> [5] Opencity: Open spatio-temporal foundation models for traffic prediction. ArXiv, 2024.
>
>
> ---
> Thank you again for your valuable feedback. We will appropriately incorporate these detailed discussions into the final revised version. We are humbled and grateful for your recognition of our work!
>
> Best regards,
>
> Authors.

---

> > ### Comment · Reviewer_N98K · 2025-08-04
> > **My concerns are addressed in the rebuttal, Please add the overview diagram in the final draft**
> >
> > The rebuttal clarifies my concerns with this paper. Yes, please add an overview diagram in the final draft. I would suggest incorporating testing results on Spatio-Temporal Foundation Models in the appendix. Thanks!

---

> > > ### Author Response · Authors · 2025-08-05
> > > **Thank you for your time and consideration**
> > >
> > > Dear Reviewer N98K,
> > >
> > > We are pleased that our response addressed all of your concerns. We will continue to improve the revised version based on your comments. We sincerely thank you for taking the time to review our paper and provide detailed and valuable feedback!
> > >
> > > Best,
> > > Authors

---

### Official Review · Reviewer_eeBu · 2025-07-03

**Clarity:** 4
**Significance:** 3
**Originality:** 2
**Rating:** 5
**Confidence:** 3

**Summary:**

Addressing distribution shift in spatio-temporal data requires some form of adaptation at test time since we rarely see all possible changes during training stages. The work thus proposes ST-TTC, which calibrates the model output in the frequency space through a learned parameter-parsimonious transformation. The adaptation module is continuously updated during inference. The experiments confirm the method's effectiveness and efficiency.

**Questions:**

Q1: Regarding l. 351, "fewer groups generally lead to poorer performance": Fig. 7 (middle) shows rather constant behaviour for varying $m$, much like $n$ in Fig. 7 (right). How was the cited statement derived from those figures?

**Ethical Concerns:**

["NO or VERY MINOR ethics concerns only"]

**Final Justification:**

The rebuttal was very convincing. I keep my "5: Accept".

The statistical significance testing is still missing. The AC should verify that them being present in the camera-ready version as promised by the authors.

**Limitations:**

Nothing is provided in the main text, which is unfortunate. However, Appendix F.1 discusses limitations in great detail.

**Paper Formatting Concerns:**

Minor: Fig. 1 (b): The second of the two closing parentheses should be black again.

**Quality:**

3

**Strengths And Weaknesses:**

**Strengths:**
- The writing is of very high quality and clarity, with exceptional Sections 4 & 5.
- The problem is significant, and the solution is not very original, yet possibly novel.
- The method is conceptually simple.
- While I am not sufficiently familiar with the field to judge the overall novelty or the completeness of datasets and baselines, their volume appears rather comprehensive.

**Opportunities for improvement (weaknesses):**
- It is unclear how the offsets $\lambda^\alpha$ and $\lambda^\phi$ are initialized and learned before inference time from the main text. Pages 21-22 clarify this, but it should be stated more clearly that there is no training stage in the main paper.
- Regarding ll. 287ff: The long-term setting is set up for only 24-to-24 time steps. However, recently, long-term time series forecasting often looks at hundreds-to-hundreds (see, e.g., Wu et al. (2022) and many more recent works). Possibly, this is substantially different in spatio-temporal forecasting.
- Regarding Thm. 1 in App. B.1: The theorem claims an exact upper bound, but the proof relies on a first-order Taylor expansion, which only provides an approximate result. To be rigorous, the theorem should either state that the bound is approximate or the proof should account for the higher-order terms.
- Checklist item 7 is answered with "yes", yet the work only provides variance measurements and no statistical tests.

**Minor Comments:**
- Fig. 3 (left): Why are the lines connected across datasets? This suggests a connection that is not there. A bar plot might be more appropriate.
- L. 366: "noval" -> "novel"
- Ll. 874-886 are a summary of the entire work and are misplaced there. They should get deleted.
- Regarding Thm. 1 in App. B.1: The result gets weakened in the very last step (last "$\leq$") of the proof, making the bounds less tight than possible. In fact, regarding Remark 1, the bounds are actually sub-linear in the modulation parameters.

**References:**
- Wu, H., Xu, J., Wang, J., & Long, M. (2022). Autoformer: Decomposition transformers with auto-correlation for long-term series forecasting. arXiv. https://doi.org/10.48550/arXiv.2106.13008

---

> ### Author Rebuttal · Authors · 2025-07-31
>
> Dear Reviewer eeBu,
>
> We sincerely appreciate the time and effort you dedicated to providing insightful feedback on our manuscript. We are honored that you recognized our hard work. We have carefully considered each of your comments and addressed them point by point, categorized by different type.
>
> ---
>
> > ### A. Method-related
>
> - **`A.1: Initialization of Offset Parameter:`**
>     - We acknowledge the lack of explicit detail regarding the initialization of the offset parameter in the main text. As you correctly pointed out, in Algorithm on page 21, both $\lambda_\alpha$ and $\lambda_\phi$ are clearly initialized to 0. The underlying motivation for this is to unify the code design of Algorithm 3, thereby preventing erroneous calibration of predictions before the calibration parameters are learned (i.e., during a very short warm-up period before acquiring the first sample-label pair). We apologize for any potential confusion and will incorporate these details into the main text.
> - **`A.2: Distinction between Long-Term Time Series and Spatio-Temporal Forecasting`**
>     - As stated in the paper, our work adheres to the common settings of previous short-term and long-term spatio-temporal forecasting studies (e.g., 12 → 12, 24 → 24) [1, 2]. However, we are fully aware of the 96 → 96~720 settings prevalent in current long-term time series forecasting. We wish to share our insights regarding this:
>         - There has been a long-standing debate concerning the long-term predictability of time series [3], which we do not intend to overly critique here. Nevertheless, it is important to note that in spatio-temporal forecasting, specifically in traffic flow theory research, previous studies [4] utilizing residual analysis have indicated that, after minimizing periodicity, the correlation between traffic data beyond one hour and the past hour's observations is significantly limited for most sensors. Furthermore, typical traffic sensor data collection frequency is approximately 5 minutes. Therefore, for practical traffic forecasting and decision-making scenarios, which usually focus on the next 1-2 hours (i.e., max 24 steps), our settings are more aligned with real-world applications.
>         - Given that spatio-temporal forecasting can be considered a complex extension of time series forecasting, the additional dimension of sensor count (up to hundreds or thousands) results in an order of magnitude higher data training cost. This is a secondary reason why existing spatio-temporal forecasting often considers 12-step settings.
>         - Despite this, we have included an experimental performance comparison for 96 → 96 prediction on the PeMS08 dataset, using STID as the backbone network, as shown below:
>         - The results show that our approach can also achieve certain performance improvements in long-term forecasting. We sincerely hope that these insights can help you better correct misunderstandings.
>
> | Model | MAE | RMSE | MAPE(%) |
> | --- | --- | --- | -- |
> | w/o ST-TTC| 16.84 | 26.80 | 11.59 |
> | w/ ST-TTC | 16.68 | 26.52 | 11.50 |
>
> ---
>
> > ### B. Presentation-related
>
> - **`B.1: Figure 3 (Left) Visualization:`**
>     - You are correct that a bar chart is more appropriate for Figure 3 (Left). We apologize for this oversight; the revised version will feature a bar chart.
> - **`B.2: Discussion of Limitations in main text:`**
>     - We will adopt your suggestion and briefly outline potential limitations in the conclusion section.
> - **`B.3: Typo Errors in Line 336 and Figure 1b:`**
>     - Thank you for your meticulous review. We will conduct a thorough check for all potential typo errors, and rectify them accordingly.
>
> ---
>
> > ### C. Experiments-related
>
> - **`C.1: The statement "fewer groups generally lead to poorer performance" on line 351 caused confusion. Figure 7 (middle) has a constant behavior similar to Figure 7 (right).`**
>     - We apologize for the confusion caused. This visual discrepancy is due to the differing metric scales between Figure 7 (Middle) and Figure 7 (Right). In actuality, the significant changes in learning rate overshadowed the impact of varying group numbers. However, performance indeed deteriorates as the number of groups decreases. Below we provide a specific example with a learning rate of 1e-3. We will avoid the above misunderstandings in the revised version.
>
> | #Groups | MAPE(%) |
> | --- | --- |
> | 1 | 24.93 |
> | 2 | 24.92 |
> | 4 | 24.85 |
> | 7 | 24.85 |
>
> - **`C.2: Statistical Significance Testing`**
>     - Thank you for your rigorousness. We plan to include statistical significance test results in the revised version.
>
> ---
>
> > ### D. Theoretical-related
>
> - **`D.1: Theorem 1 should be stated as an approximate result and is, strictly speaking, sub-linear.`**
>     - We appreciate your precision. We will add more detailed supplements and discussions in the revised version.
>
> ---
>
> **References:**
>
> [1] Zezhi Shao, et al. Exploring Progress in Multivariate Time Series Forecasting: Comprehensive Benchmarking and Heterogeneity Analysis. TKDE, 2024.
>
> [2] Wei Shao, et al. Long-term spatio-temporal forecasting via dynamic multiple-graph attention. IJCAI, 2022.
>
> [3] Christoph Bergmeir. Fundamental limitations of foundational forecasting models. NeurIPS Workshop InvTalk, 2024.
>
> [4] Lorenzo Brigato, et al. Position: There are no Champions in Long-Term Time Series Forecasting. arXiv, 2025.
>
> [5] Vincent Zhihao Zheng, et al. Probabilistic Traffic Forecasting with Dynamic Regression. Transportation Science, 2025.
>
> ---
>
> Thank you for your invaluable feedback. We will appropriately incorporate these detailed discussions into the final revised version. Once again, we are grateful for your insightful guidance!
>
> Best regards,
>
> Authors.

---

> > ### Comment · Reviewer_eeBu · 2025-08-04
> >
> > Thank you very much for your detailed response. It convinced me to maintain my score of "5: Accept".
> >
> > I agree with the take on what is or should be long-term. The problem lies with the terminology in the community, not the evaluation performed.
> >
> > The statistical significance testing is still missing and should certainly be added in the camera-ready version of the work.

---

> > > ### Author Response · Authors · 2025-08-05
> > > **Thank you for your time and consideration**
> > >
> > > Dear Reviewer eeBu,
> > >
> > > We are glad to hear that our rebuttal effectively addressed your concerns. We will improve the revised version based on your feedback. Thank you again for taking the time and effort to provide valuable feedback on our paper!
> > >
> > > Best,
> > >
> > > Authors

---

### Note · Authors · 2025-08-12

Dear AC and Reviewers,

We sincerely thank AC and all the reviewers for your time and effort. **It is encouraging to see that almost every reviewer has recognized the positive aspects of our manuscript**, such as: `important and practical problem` (Reviewer eeBu, xiA6), `promising research direction` (Reviewer xiA6, T8C2), `well-motivated design choices` (Reviewer N98K, xiA6), `a novel and scalable solution` (Reviewer eeBu, N98K, xiA6, T8C2), `solid empirical analyses` (Reviewer eeBu, N98K, xiA6, T8C2), `robust performance gains` (reviewer N98K, xiA6, T8C2), and `high-quality, clear presentation` (Reviewer eeBu, N98K).

We have provided detailed responses to each reviewer and addressed almost all raised concerns. In this general response, we summarize the planned revisions for the final version, following reviewers’ valuable suggestions. We hope these responses adequately address any potential concerns from the reviewers and AC.

- **Presentation** (per eeBu, N98K): ① revise the Fig. 3 left to a bar chart; ② correct all typos; ③ add an overview pipeline diagram.
- **Methodology** (per eeBu): ① add more details on parameter initialization; ② enhance precision and completeness of theoretical claim.
- **Experiments** (per eeBu, N98K): ① include statistical significance tests; ② add evaluation on spatio-temporal foundation models; ③ provide additional quantitative visualizations for OOD datasets; ④ add evaluation on video prediction tasks.
- **Discussion** (per eeBu, xiA6, T8C2): ① expand discussion on long- vs. short-term forecasting; ② discuss generalization to anomalies and sudden structural changes.

It is noteworthy that the above revisions were nearly completed during the discussion phase, and the reviewers’ expertise has been invaluable in strengthening our manuscript. **We once again sincerely thank all reviewers for their recognition and constructive feedback, as well as the AC for the thorough review.**

Sincerely,

Authors of Submission 6767

---

### Decision · Program_Chairs · 2025-09-17

**Decision:**

Accept (spotlight)

**Comment:**

This paper focuses on distributional shifts in spatio-temporal data. To address this issue, the authors propose a novel test-time computing paradigm, termed learning with calibration (ST-TTC), which aims to capture periodic structural biases induced by non-stationarity during the testing phase. Overall, the work is well-presented and solid in both technical quality and experimental evaluation. All reviewers reached a positive consensus after the author–reviewer discussion stage.